# Restricted Category Removal from Model Representations using Limited Data

## Abstract

Deep learning models are trained on multiple categories jointly to solve several real-world problems. However, there can be cases where some of the classes may become restricted in the future and need to be excluded after the model has already been trained on them (Class-level Privacy). It can be due to privacy, ethical or legal concerns. A naive solution is to simply train the model from scratch on the complete training data while leaving out the training samples from the restricted classes (FDR - full data retraining). But this can be a very time-consuming process. Further, this approach will not work well if we no longer have access to the complete training data and instead only have access to very few training data. The objective of this work is to remove the information about the restricted classes from the network representations of all layers using limited data without affecting the prediction power of the model for the remaining classes. Simply fine-tuning the model on the limited available training data for the remaining classes will not be able to sufficiently remove the restricted class information, and aggressive fine-tuning on the limited data may also lead to overfitting. We propose a novel solution to achieve this objective that is significantly faster ($\sim 200\times$ on ImageNet) than the naive solution. Specifically, we propose a novel technique for identifying the model parameters that are mainly relevant to the restricted classes. We also propose a novel technique that uses the limited training data of the restricted classes to remove the restricted class information from these parameters and uses the limited training data of the remaining classes to reuse these parameters for the remaining classes. The model obtained through our approach behaves as if it was never trained on the restricted classes and performs similar to FDR (which needs the complete training data). We also propose several baseline approaches and compare our approach with them in order to demonstrate its efficacy.

## 1 Introduction

There are several real-world problems in which deep learning models have exceeded human-level performance. This has led to a wide deployment of deep learning models. Deep learning models generally train jointly on a number of categories/classes of data. However, the use of some of these classes may get restricted in the future (restricted classes), and a model with the capability to identify these classes may violate legal/privacy concerns, e.g., a company may legally prevent a deep learning model from having the capability to identify its copyright-protected logo, patented products, and so on. Another example is a treatment prediction model that predicts the best treatment for a patient based on the disease. If one of the treatments for a disease is banned due to its side-effects or ethical concerns, the restricted treatment category has to be excluded from the trained model. Individuals and organizations are becoming increasingly aware of these issues leading to an increasing number of legal cases on privacy issues in recent years. In such situations, the model has to be stripped of its capability to identify these categories. This is a difficult problem to solve, especially if the full training data is no longer available and only a few training examples are available. We present a "Restricted Category Removal from Model Representations with Limited Data" (RCRMR-LD) problem setting that simulates the above problem. In this paper, we propose to solve this problem in a fast and efficient manner.

The objective of the RCRMR-LD problem is to remove the information regarding the restricted classes from the network representations of all layers using the limited training data available without

affecting the ability of the model to identify the remaining classes. If we have access to the full training data, then we can simply exclude the restricted class examples from the training data and perform a full training of the model from scratch using the abundant data (FDR - full data retraining). However, the RCRMR-LD problem setting is based on the scenario that the directive to exclude the restricted classes is received in the future after the model has already been trained on the full data and now only a limited amount of training data is available to carry out this process. Simply training the network from scratch on only the limited training data of the remaining classes will result in severe overfitting and significantly affect the model performance (Baseline 2, as shown in Tables 1, 6).

Another possible solution to this problem is to remove the weights of the fully-connected classification layer of the network corresponding to the excluded classes such that it can no longer classify the excluded classes. However, this approach suffers from a serious problem. Since, in this approach, we only remove some of the weights of the classification layer and the rest of the model remains unchanged, it still contains the information required for recognizing the excluded classes. This information can be easily accessed through the features that the model extracts from the images. Therefore, we can use these features for performing classification. In this paper, we use a nearest prototype-based classifier to demonstrate that the model features still contain information regarding the restricted classes. Specifically, we use the model features of the examples from the limited training data to compute the average class prototype for each class and create a nearest class prototype-based classifier using them. Next, for any given test image, we extract its features using the model and then find the class prototype closest to the given test image. This nearest class prototype-based classifier performs close to the original fully-connected classifier on the excluded classes as shown in Tables 1, 6 (Baseline 1). Therefore, even after using this approach, the resulting model still contains information regarding the restricted classes. Another possible approach can be to apply the standard fine-tuning approach to the model using the limited available training data of the remaining classes (Baseline 8). However, fine-tuning on such limited training data is not able to sufficiently remove the restricted class information from the model representations (see Tables 1, 6), and aggressive fine-tuning on the limited training data may result in overfitting.

Considering the problems faced by the naive approaches mentioned above, we propose a novel "Efficient Removal with Preservation" (ERwP) approach to address the RCRMR-LD problem. First, we propose a novel technique to identify the model parameters that are highly relevant to the restricted classes, and to the best of our knowledge, there are no existing prior works for finding such class-specific relevant parameters. Next, we propose a novel technique that optimizes the model on the limited available training data in such a way that the restricted class information is discarded from the restricted class relevant parameters, and these parameters are reused for the remaining classes.

To the best of our knowledge, this is the first work that addresses the RCRMR-LD problem. Therefore, we also propose several baseline approaches for this problem (see Sec. 11.2). However, our proposed approach significantly outperforms all the proposed baseline approaches. Our proposed approach requires very few epochs to address the RCRMR-LD problem and is, therefore, very fast ($\sim 200$ times faster than the full data retraining model for the ImageNet dataset) and efficient. The model obtained after applying our approach forgets the excluded classes to such an extent that it behaves as though it was never trained on examples from the excluded classes. The performance of our model is very similar to the full data retraining (FDR) model (see Sec. 8.1, Fig. 5). We also propose the performance metrics needed to evaluate the performance of any approach for the RCRMR-LD problem. We perform experiments on several datasets to demonstrate the efficacy of our method.

## 2 PROBLEM SETTING

In this work, we present the *restricted category removal from model representations with limited data (RCRMR-LD)* problem setting, in which a deep learning model $M_o$ trained on a specific dataset has to be modified to exclude information regarding a set of restricted/excluded classes from all layers of the deep learning model without affecting its identification power for the remaining classes (see Fig. 1). The classes that need to be excluded are referred to as the restricted/excluded classes. Let $\{C_1^e, C_2^e, ..., C_{N_e}^e\}$ be the restricted/excluded classes, where $N_e$ refers to the number of excluded classes. The remaining classes of the dataset are the remaining/non-excluded classes. Let $\{C_1^{ne}, C_2^{ne}, ..., C_{N_{ne}}^{ne}\}$ be the non-excluded classes, where $N_{ne}$ refers to the number of remaining/non-excluded classes. Additionally, we only have access to a limited amount of training data for the restricted classes and the remaining classes, for carrying out this process. Therefore, any approach for addressing this problem can only utilize this limited training data.

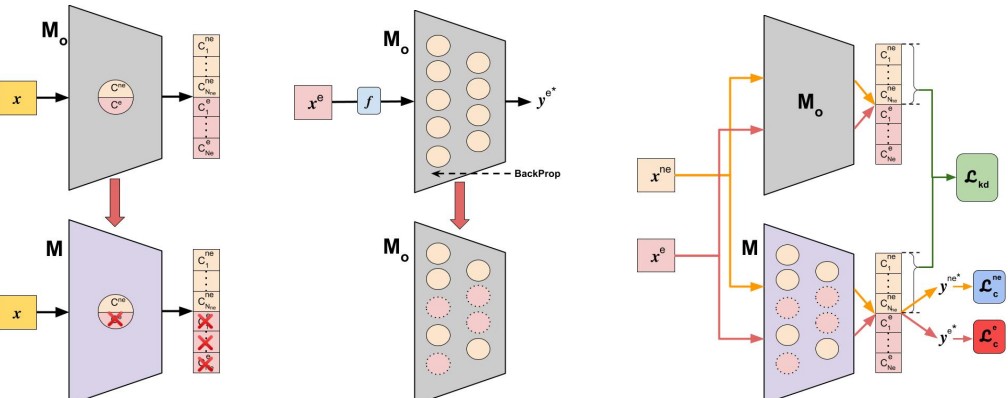

Figure 1: The RCRMR-LD problem setting aims to remove the information regarding the restricted/excluded classes ($\{C_1^e, .., C_{N_e}^e\}$) from all layers of a trained model $M_o$ while preserving its predictive power for the remaining classes ($\{C_1^{ne}, .., C_{N_{ne}}^{ne}\}$) using limited training data. The category removal (denoted by a red cross) has to take place at the classifier level (denoted as squares for each output logit) and at the feature/representation level (denoted as a circle).

Figure 2: ERwP identifies those parameters in the model that are highly relevant to the restricted classes. To obtain these parameters, ERwP modifies training images from a restricted class using a data augmentation $f$ and performs backpropagation using the classification loss on these training images. ERwP then studies the gradient update that each parameter receives in this process in order to identify the highly relevant parameters for the restricted classes (denoted by dotted circles).

Figure 3: ERwP only optimizes the restricted class relevant parameters in the model (denoted by dotted circles). ERwP uses $\mathcal{L}_c^e$, $\mathcal{L}_c^{ne}$ and $\mathcal{L}_{kd}$ losses to remove the restricted class information from the model while preserving its performance on the remaining classes. $\mathcal{L}_c^e$ and $\mathcal{L}_c^{ne}$ denote the classification loss on the restricted class training examples and the remaining class training example, respectively. $\mathcal{L}_{kd}$ denotes the knowledge distillation-based regularization loss that preserves the logits corresponding to only the remaining classes for all the training examples.

## 3 RCRMR-LD PROBLEM IN REAL WORLD SCENARIOS

A real-world scenario where our proposed RCRMR-LD problem can arise is federated learning (McMahan et al., 2017). In the federated learning setting, there are multiple collaborators that have a part of the training data stored locally, and a model is trained collaboratively using these private data without sharing or collating the data due to privacy concerns. Suppose organization A has a part of the training data, and there are other collaborators that have other parts of the training data for the same classes. Organization A collaboratively trains a model with other collaborators using federated learning. After the model has been trained, a few classes may become restricted in the future due to some ethical or privacy concerns, and these classes should be removed from the model. However, the other collaborators may not be available or may charge a huge amount of money for collaborating again to train a fresh model from scratch. In this case, organization A does not have access to the full training data of the non-excluded/remaining classes that it can use to re-train a model from scratch in order to exclude the restricted classes information. This clearly shows that the RCRMR-LD problem is possible in federated learning.

Another real-world scenario is the incremental learning setting (Rebuffi et al., 2017; Kemker & Kanan, 2018), where the model receives training data in the form of sequentially arriving tasks. Each task contains a new set of classes. During a training session $t$, the model receives the task $t$ for training and cannot access the full data of the previous tasks. Instead, the model has access to very few exemplars of the classes in the previous tasks. Suppose before training a model on training session $t$, it is noticed that some classes from a previous task ($< t$) have to be removed from the model since those classes have become restricted due to privacy or ethical concerns. In this case, only a limited number of exemplars are available for all these previous classes (restricted and remaining). This demonstrates that the RCRMR-LD problem is also possible in the incremental learning setting. We experimentally demonstrate in Sec. 8.3, how our approach can address the RCRMR-LD problem in the incremental learning setting.

## 4   PROPOSED METHOD

Let, $B$ refer to a mini-batch (of size $S$) from the available limited training data, and $B$ contains training datapoints from the restricted/excluded classes ($\{(x_i^e, y_i^e)|(x_1^e, y_1^e), ..., (x_{S_e}^e, y_{S_e}^e)\}$) and from the remaining/non-excluded classes ($\{(x_j^{ne}, y_j^{ne})|(x_1^{ne}, y_1^{ne}), ..., (x_{S_{ne}}^{ne}, y_{S_{ne}}^{ne})\}$). Here, $(x_i^e, y_i^e)$ refers to a training datapoint from the excluded classes where $x_i^e$ is an image, $y_i^e$ is the corresponding label and $y_i^e \in \{C_1^e, C_2^e, ..., C_{N_e}^e\}$. $(x_j^{ne}, y_j^{ne})$ refers to a training datapoint from the non-excluded classes where $x_j^{ne}$ is an image, $y_j^{ne}$ is the corresponding label and $y_j^{ne} \in \{C_1^{ne}, C_2^{ne}, ..., C_{N_{ne}}^{ne}\}$. Here, $S_e$ and $S_{ne}$ refer to the number of training examples in the mini-batch from the excluded and non-excluded classes, respectively, such that $S = S_e + S_{ne}$. $N_e$ and $N_{ne}$ refer to the number of excluded and non-excluded classes, respectively. Let $M$ refer to the deep learning model being trained using our approach and $M_o$ is the original trained deep learning model.

In a trained model, some of the parameters may be highly relevant to the restricted classes, and the performance of the model on the restricted classes is mainly dependent on such highly relevant parameters. Therefore, in our approach, we focus on removing the excluded class information from these restricted class relevant parameters. Since the model is trained on all the classes jointly, the parameters are shared across the different classes. Therefore identifying these class-specific relevant parameters is very difficult. Let us consider a model that is trained on color images of a class. If we now train it on grayscale images of the class, then the model has to learn to identify these new images. In order to do so, the parameters relevant to that class will receive large gradient updates as compared to the other parameters (see Sec. 9.1). We propose a novel approach for identifying the relevant parameters for the restricted classes using this idea. For each restricted class, we choose the training images belonging to that class from the limited available training data. Next, we apply a grayscale data augmentation technique/transformation $f$ to these images so that these images become different from the images that the original model was earlier trained on (assuming that the original model has not been trained on grayscale images). We can also use other data augmentation techniques that are not seen during the training process of the original model and that do not change the class of the image (refer to Sec. 11.7 in the appendix). Next, we combine the predictions for each training image into a single average prediction and perform backpropagation. During the backpropagation, we study the gradients for all the parameters in each layer of the model. Accordingly, we select the parameters with the highest absolute gradient as the relevant parameters for the corresponding restricted class. Specifically, for a given restricted class, we choose the minimum number of such parameters from each network layer such that pruning these parameters will result in the maximum degradation of model performance on that restricted class. We provide a detailed description of the process for identifying the restricted class relevant parameters in Sec. 11.1 of the appendix. The combined set of the relevant parameters for all the excluded classes is referred to as the restricted/excluded class relevant parameters $\Theta_{exrel}$ (see Fig. 2). Please note that we use this process only to identify $\Theta_{exrel}$, and we do not update the model parameters during this step.

Pruning the relevant parameters for a restricted class can severely impact the performance of the model for that class (see Sec. 9.1). However, this may also degrade the performance of the model on the non-excluded classes because the parameters are shared across multiple classes. Therefore, we cannot address the RCRMR-LD problem by pruning the relevant parameters of the excluded classes. Finetuning these parameters on the limited remaining class data will also not be able to sufficiently remove the restricted class information from the model (see Sec. 11.10 in appendix). Based on this, we propose to address the RCRMR-LD problem by optimizing the relevant parameters of the restricted classes to remove the restricted class information from them and to reuse them for the remaining classes.

After identifying the restricted class relevant parameters, our ERwP approach uses a classification loss based on the cross-entropy loss function to optimize the restricted class relevant parameters of the model on each mini-batch (see Fig. 3). We know that the gradient ascent optimization algorithm can be used to maximize a loss function and encourage the model to perform badly on the given input. Therefore, we use the gradient ascent optimization on the classification loss for the limited restricted class training examples to remove the information regarding the restricted classes from $\Theta_{exrel}$. We achieve this by multiplying the classification loss for the augmented training examples from the excluded classes by a constant negative factor of -1. We also optimize $\Theta_{exrel}$ using the gradient descent optimization on the classification loss for the limited remaining class training example, in order to reuse these parameters for the remaining classes. We validate using this approach through various ablation experiments as shown in Sec. 9.2. The classification loss for the examples from the

excluded and non-excluded classes and the overall classification loss for each mini-batch are defined as follows.

$$\mathcal{L}_c^e = \sum_{i=1}^{S_e} -1 * \ell(y_i^e, y_i^{e*}) \tag{1}$$

$$\mathcal{L}_c^{ne} = \sum_{j=1}^{S_{ne}} \ell(y_j^{ne}, y_j^{ne*}) \tag{2}$$

$$\mathcal{L}_c = \frac{1}{S}(\mathcal{L}_c^e + \mathcal{L}_c^{ne}) \tag{3}$$

Where, $y_i^{e*}$ and $y_j^{ne*}$ refer to the predicted class labels for $x_i^e$ and $x_j^{ne}$, respectively. $\ell(.,.)$ refers to the cross-entropy loss function. $\mathcal{L}_c^e$ and $\mathcal{L}_c^{ne}$ refer to the classification loss for the examples from the excluded and non-excluded classes in the mini-batch, respectively. $\mathcal{L}_c$ refers to the overall classification loss for each mini-batch.

Since all the network parameters were jointly trained on all the classes (restricted and remaining), the restricted class relevant parameters also contain information relevant to the remaining classes. Applying the above process alone will still harm the model's predictive power for the non-excluded classes (as shown in Sec. 9.2, Table 3). This is because the gradient ascent optimization strategy will also erase some of the relevant information regarding the remaining classes. Further, applying $\mathcal{L}_c^{ne}$ on the limited training examples of the remaining classes will lead to overfitting and will not be effective enough to fully preserve the model performance on the remaining classes. In order to ensure that the model's predictive power for the non-excluded classes does not change, we use a knowledge distillation-based regularization loss. Knowledge distillation (Hinton et al., 2014) ensures that the predictive power of the teacher network is replicated in the student network. In this problem setting, we want the final model to replicate the same predictive power of the original model for the remaining classes. Therefore, given any training example, we use the knowledge distillation-based regularization loss to ensure that the output logits produced by the model corresponding to only the non-excluded classes remain the same as that produced by the original model. We apply the knowledge distillation loss to the limited training examples from both the excluded and remaining classes, to preserve the non-excluded class logits of the model for any input image. We validate this regularization loss through ablation experiments as shown in Table 3. We use the original model $M_o$ (before applying ERwP) as the teacher network and the current model $M$ being processed by ERwP as the student network, for the knowledge distillation process. Please note that the optimization for this loss is also carried out only for the restricted class relevant parameters of the model. Let $KD$ refer to the knowledge distillation loss function. It computes the Kullback-Liebler (KL) divergence between the soft predictions of the teacher and the student networks and can be defined as follows:

$$KD(p_s, p_t) = KL(\sigma(p_s), \sigma(p_t)) \tag{4}$$

where, $\sigma(.)$ refers to the softmax activation function that converts logit $a_i$ for each class $i$ into a probability by comparing $a_i$ with logits of other classes $a_j$, i.e., $\sigma(a_i) = \frac{exp^{a_i/\kappa}}{\sum_j exp^{a_j/\kappa}}$. $\kappa$ refers to the temperature (Hinton et al., 2014), $KL$ refers to the KL-Divergence function. $p_s, p_t$ refer to the logits produced by the student network and the teacher network, respectively.

The knowledge distillation-based regularization losses in our approach are defined as follows.

$$\mathcal{L}_{kd}^e = \sum_{i=1}^{S_e} KD(M(x_i^e)[C^{ne}], M_o(x_i^e)[C^{ne}]) \tag{5}$$

$$\mathcal{L}_{kd}^{ne} = \sum_{j=1}^{S_{ne}} KD(M(x_j^{ne})[C^{ne}], M_o(x_j^{ne})[C^{ne}]) \tag{6}$$

$$\mathcal{L}_{kd} = \frac{1}{S}(\mathcal{L}_{kd}^e + \mathcal{L}_{kd}^{ne}) \tag{7}$$

Where, $M(\#)[C^{ne}]$ and $M_o(\#)[C^{ne}]$ refer to the output logits corresponding to the remaining classes produced by $M$ and $M_o$, respectively. $\#$ can be either $x_i^e$ or $x_j^{ne}$. $\mathcal{L}_{kd}^e$ and $\mathcal{L}_{kd}^{ne}$ refer to knowledge distillation-based regularization loss for the examples from the excluded and non-excluded classes, respectively. $\mathcal{L}_{kd}$ refers to the overall knowledge distillation-based regularization loss for each mini-batch.

The total loss $\mathcal{L}_{erwp}$ of our approach for each mini-batch is defined as follows.

$$\mathcal{L}_{erwp} = \mathcal{L}_c + \beta\mathcal{L}_{kd} \tag{8}$$

Where, $\beta$ is a hyper-parameter that controls the contribution of the knowledge distillation-based regularization loss. We use this loss for fine-tuning the model for very few epochs.

## 5   RELATED WORK

Pruning involves removing redundant and unimportant weights (Carreira-Perpinán & Idelbayev, 2018; Dong et al., 2017; Guo et al., 2016; Han et al., 2015a;b; Tung & Mori, 2018; Zhang et al., 2018) or filters (He et al., 2019a; 2018; 2019b;c; Li et al., 2016) from a deep learning model without affecting the model performance. In contrast, our approach identifies class-specific important parameters, and therefore, the pruning techniques cannot be applied in our approach. In the incremental learning setting (Douillard et al., 2020; Hou et al., 2019; Tao et al., 2020; Yu et al., 2020; Liu et al., 2021), the objective is to preserve the predictive power of the model for previously seen classes while learning a new set of classes. In contrast, our proposed RCRMR-LD problem setting involves removing the information regarding specific classes from the pre-trained model while preserving the capacity of the model to identify the remaining classes. Privacy-preserving deep learning (Nan & Tao, 2020; Louizos et al., 2015; Edwards & Storkey, 2015; Hamm, 2017) involves learning representations that incorporate features from the data relevant to the given task and ignore sensitive information (such as the identity of a person). In contrast, the objective of the RCRMR-LD problem setting, is to achieve class-level privacy, i.e., if a class is declared as private/restricted, then all information about this class should be removed from the model trained on it, without affecting its ability to identify the remaining classes. The authors in (Ginart et al., 2019) propose an approach to delete individual data points from trained machine learning models like a clustering model. In contrast, RCRMR-LD involves removing the information of a set of classes from all layers of a deep learning model. Therefore, the approach proposed in (Ginart et al., 2019) cannot be applied to the RCRMR-LD problem setting.

## 6   BASELINES

We propose 9 baseline models for the RCRMR-LD problem and compare our proposed approach with them. The baseline 1 involves deleting the weights of the fully-connected classification layer corresponding to the excluded classes. Baselines 2, 3, 4, 5 involve training the model on the limited training data of the remaining classes. Baselines 6, 7, 8, 9 involve fine-tuning the model on the available limited training data. Please refer to Sec. 11.2 in the appendix for details about the baselines.

## 7   PERFORMANCE METRICS

In the RCRMR-LD problem setting, we propose three performance metrics to validate the performance of any method: forgetting accuracy ($\text{FA}_e$), forgetting prototype accuracy ($\text{FPA}_e$), and constraint accuracy ($\text{CA}_{ne}$). The forgetting accuracy refers to the fully-connected classification layer accuracy of the model for the excluded classes. The forgetting prototype accuracy refers to the nearest class prototype-based classifier accuracy of the model for the excluded classes. $\text{CA}_{ne}$ refers to the fully-connected classification layer accuracy of the model for the non-excluded classes.

In order to judge any approach on the basis of these metrics, we follow the following sequence. First, we analyze the constraint accuracy ($\text{CA}_{ne}$) of the model produced by the given approach to verify if the approach has preserved the prediction power of the model for the non-excluded classes. $\text{CA}_{ne}$ of the model should be close to that of the original model. If this condition is not satisfied, then the approach is not suitable for this problem, and we need not analyze the other metrics. This is because if the constraint accuracy is not maintained, then the overall usability of the model is hurt significantly. Next, we analyze the forgetting accuracy ($\text{FA}_e$) of the model to verify if the excluded class information has been removed from the model at the classifier level. $\text{FA}_e$ of the model should be as close to 0% as possible. Finally, we analyze the forgetting prototype accuracy ($\text{FPA}_e$) of the model to verify if the excluded class information has been removed from the model at the feature level. $\text{FPA}_e$ of the model should be significantly less than that of the original model. However, the $\text{FPA}_e$ will not become zero since any trained model will learn to extract meaningful features, which will help the nearest class prototype-based classifier to achieve some non-negligible accuracy even on the excluded classes. Therefore, for a better analysis of the level of forgetting of the excluded classes at the feature level, we compare the $\text{FPA}_e$ of the model with the $\text{FPA}_e$ of the FDR model. The

Table 1: Experimental results on the CIFAR-100 dataset for the RCRMR-LD problem.

| Methods | ResNet-20 | | | ResNet-56 | | | ResNet-164 | | |
|---|---|---|---|---|---|---|---|---|---|
| | $FA_e$ | $FPA_e$ | $CA_{ne}$ | $FA_e$ | $FPA_e$ | $CA_{ne}$ | $FA_e$ | $FPA_e$ | $CA_{ne}$ |
| Original | 70.15% | 65.25% | 67.06% | 70.80% | 68.65% | 69.88% | 79.00% | 76.40% | 76.30% |
| **No Training** | | | | | | | | | |
| Baseline 1 - WD | 0.00% | 65.25% | 69.88% | 0.00% | 68.65% | 72.44% | 0.00% | 76.40% | 77.01% |
| **Full Train Schedule** | | | | | | | | | |
| Baseline 2 - TSLNRC | 0.00% | 22.20% | 31.55% | 0.00% | 20.20% | 30.21% | 0.00% | 33.05% | 40.65% |
| Baseline 3 - TSLNRC-KD | 0.00% | 27.55% | 40.81% | 0.00% | 22.50% | 32.26% | 0.00% | 38.55% | 45.74% |
| Baseline 4 - TOLNRC | 0.00% | 50.85% | 58.01% | 0.00% | 48.60% | 57.81% | 0.00% | 51.55% | 62.78% |
| Baseline 5 - TOLNRC-KD | 0.00% | 60.25% | 67.85% | 0.00% | 51.25% | 61.14% | 0.00% | 52.80% | 63.75% |
| **Only Fine-Tuning** | | | | | | | | | |
| Baseline 6 - FOLMRCSC | 24.25% | 59.55% | 64.03% | 13.35% | 60.25% | 65.23% | 15.40% | 59.20% | 71.06% |
| Baseline 7 - FOLMRCSC-KD | 13.50% | 58.80% | 63.79% | 12.75% | 64.95% | 63.41% | 16.75% | 65.30% | 68.61% |
| Baseline 8 - FOLNRC | 59.05% | 64.30% | 68.34% | 66.90% | 68.45% | 70.11% | 77.35% | 75.85% | 75.95% |
| Baseline 9 - FOLNRC-KD | 57.99% | 64.40% | 68.40% | 65.95% | 68.40% | 70.01% | 73.30% | 73.55% | 75.99% |
| **ERwP (Ours)** | 0.00% | 48.06% | 66.84% | 0.00% | 47.84% | 69.32% | 0.74% | 56.23% | 75.65% |

FDR model is a good candidate for this analysis since it has not been trained on the excluded classes (only trained on the complete dataset of the remaining classes), and it still achieves a non-negligible performance of the excluded classes (see Sec 8.1). However, it should be noted that this comparison is only for analysis and the comparison is not fair since the FDR model needs to train on the entire dataset (except the excluded classes).

# 8 EXPERIMENTS

We have reported the experimental results for the CIFAR-100 and ImageNet-1k datasets in this section. We have provided the results on the CUB-200 dataset in the appendix. Please refer to the appendix for the details regarding the dataset and implementation.

## 8.1 CIFAR-100 RESULTS

We report the performance of different baselines and our proposed ERwP method on the RCRMR-LD problem using the CIFAR-100 dataset with different architectures in Table 1. We observe that the baseline 1 (weight deletion) achieves high constraint accuracy $CA_{ne}$ and 0% forgetting accuracy $FA_e$. But its forgetting prototype accuracy $FPA_e$ remains the same as the original model for all the three architectures, i.e., ResNet-20/56/164. Therefore, baseline 1 fails to remove the excluded class information from the model at the feature level. Baseline 2 is not able to preserve the constraint accuracy $CA_{ne}$ even though it performs full training on the limited excluded class data. Baseline 3 achieves higher $CA_{ne}$ than baseline 2, but the constraint accuracy is still too low. Baselines 4 and 5 demonstrate significantly better constraint accuracy than baseline 2 and 3, but their constraint accuracy is still significantly lower than the original model (except baseline 5 for ResNet-20). The baseline 5 with ResNet-20 maintains the constraint accuracy and achieves 0% forgetting accuracy $FA_e$ but its $FPA_e$ is still significantly high and, therefore, is unable to remove the excluded class information from the model at the feature level. The fine-tuning based baselines 6 and 7 are able to significantly reduce the forgetting accuracy $FA_e$ but their constraint accuracy $CA_{ne}$ drops significantly. The fine-tuning based baselines 8 and 9 only finetune the model on the limited remaining class data and as a result they are not able to sufficiently reduce either the forgetting accuracy $FA_e$ or the forgetting prototype accuracy $FPA_e$.

Our proposed ERwP approach achieves a constraint accuracy $CA_{ne}$ that is very close to the original model for all three architectures. It achieves close to 0% $FA_e$. Further, it achieves a significantly lower $FPA_e$ than the original model. Specifically, the $FPA_e$ of our approach is significantly lower than that of the original model by absolute margins of 17.19%, 20.81%, and 20.17% for the ResNet-20, ResNet-56, and ResNet-164 architectures, respectively. The $FPA_e$ for the FDR model is 44.20%, 45.40% and 51.85% for the ResNet-20, ResNet-56 and ResNet-164 architectures, respectively. Therefore, the $FPA_e$ of our approach is close to that of the FDR model by absolute margins of 3.86%, 2.44% and 4.38% for the ResNet-20, ResNet-56 and ResNet-164 architectures, respectively. Therefore, our ERwP approach makes the model behave similar to the FDR model even though it was trained on only limited data from the excluded and remaining classes. Further, our ERwP requires only 10 epochs to remove the excluded class information from the model. Since the available limited training

Table 2: Experimental results on ImageNet-1k.

| Model | Methods | Top-1 | | Top-5 | |
|---|---|---|---|---|---|
| | | $FA_e$ | $CA_{ne}$ | $FA_e$ | $CA_{ne}$ |
| Res-18 | Original | 69.76% | 69.76% | 89.58% | 89.02% |
| | **ERwP** | 0.28% | 69.13% | 1.01% | 88.93% |
| Res-50 | Original | 76.30% | 76.11% | 93.04% | 92.84% |
| | **ERwP** | 0.25% | 75.45% | 2.55% | 92.39% |
| Mob-V2 | Original | 72.38% | 70.83% | 91.28% | 90.18% |
| | **ERwP** | 0.17% | 70.81% | 0.81% | 89.95% |

Table 3: Significance of ERwP components.

| $\mathcal{L}_c^e$ | $\mathcal{L}_c^{ne}$ | $\mathcal{L}_{kd}$ | $FA_e$ | $FPA_e$ | $CA_{ne}$ |
|---|---|---|---|---|---|
| ✓ | ✗ | ✗ | 66.50% | 68.19% | 69.79% |
| ✓ | ✓ | ✗ | 0.00% | 24.40% | 6.45% |
| ✓ | ✓ | ✓ | 0.00% | 47.84% | 69.32% |

Figure 4: Ablation to validate our approach for identifying relevant model parameters for a random restricted class of CIFAR-100.

Table 4: Performance of ERwP in the incremental learning setting using ResNet-18.

| Model | $FA_e$ | $CA_{ne}$ |
|---|---|---|
| Original Model obtained after Session 4 [M4] | 56.39% | 58.32% |
| **M4 modified with ERwP (Ours)** | 0.20% | 59.93% |

data is only 10% of the entire CIFAR-100 dataset, therefore, our ERwP approach is approximately $30 * 10 = 300\times$ faster than the FDR method that is trained on the full training data for 300 epochs.

## 8.2 IMAGENET RESULTS

Table 2 reports the experimental results for different approaches to RCRMR-LD problem over the ImageNet-1k dataset using the ResNet-18, ResNet-50 and MobileNet V2 architectures. Our proposed ERwP approach achieves a top-1 constraint accuracy $CA_{ne}$ that is very close to that of the original model by absolute margins of 0.63%, 0.66% and 0.02% for the ResNet-18, ResNet-50 and MobileNet V2 architectures, respectively. It achieves close to 0% top-1 forgetting accuracy $FA_e$ for all the three architectures. Therefore, our approach performs well even on the large-scale ImageNet-1k dataset. Further, our ERwP requires only 10 epochs to remove the excluded class information from the model. Since the available limited training data is only 5% of the entire CIFAR-100 dataset, therefore, our ERwP approach is approximately $20 * 10 = 200\times$ faster than the FDR method that is trained on the full data for 100 epochs.

## 8.3 RCRMR-LD PROBLEM IN INCREMENTAL LEARNING

In this section, we experimentally demonstrate how the RCRMR-LD problem in the incremental learning setting is addressed using our proposed approach. We consider an incremental learning setting on the CIFAR-100 dataset in which each task contains 20 classes. We use the BIC (Wu et al., 2019) method for incremental learning on this dataset. The exemplar memory size is fixed at 2000 as per the setting in (Wu et al., 2019). In this setting, there are 5 tasks. Let us assume that the model (M4) has already been trained on 4 tasks (80 classes), and we are in the fifth training session. Suppose, at this stage, it is noticed that all the classes in the first task (20 classes) have become restricted and need to be removed before the model is trained on task 5. However, we only have a limited number of exemplars of the 80 classes seen till now, i.e., 2000/80 = 25 per class. We apply our proposed approach to the model obtained after training session 4, and the results are reported in Table 4. The results indicate that our approach modified the model obtained after session 4, such that the forgetting accuracy of the restricted classes approaches 0% and the constraint accuracy of the remaining classes is not affected. In fact, the modified model behaves as if, it was never trained on the classes from task 1. We can now perform the incremental training of the modified model on task 5.

## 9 ABLATION STUDIES

### 9.1 RESTRICTED CLASS RELEVANT PARAMETERS

We perform ablation experiments to verify our approach of identifying the highly relevant parameters for any restricted class. We perform these experiments on the CIFAR-100 dataset with the ResNet-56 architecture and report the forgetting accuracy $FA_e$ for the randomly chosen excluded class. Please note that in this case, only the chosen class of CIFAR-100 is the restricted class and all the remaining

Table 5: Experimental results on the CIFAR-100 dataset using ResNet-56 for ERwP with different number of excluded classes. # R/E → no. of non-excluded classes / no. of excluded classes

| # R/E | Methods | ResNet-20 | | ResNet-56 | |
|---|---|---|---|---|---|
| | | FA$_e$ | CA$_{ne}$ | FA$_e$ | CA$_{ne}$ |
| 60/40 | Original | 68.18% | 67.35% | 69.98% | 70.11% |
| | ERwP | 0.00% | 67.03% | 0.00% | 69.98% |
| 70/30 | Original | 67.83% | 67.61% | 69.60% | 70.26% |
| | ERwP | 0.00% | 67.25% | 0.00% | 69.81% |
| 80/20 | Original | 70.15% | 67.06% | 70.80% | 69.88% |
| | ERwP | 0.00% | 66.85% | 0.00% | 69.26% |
| 90/10 | Original | 67.90% | 67.66% | 68.40% | 70.24% |
| | ERwP | 0.00% | 67.26% | 0.00% | 69.69% |
| 95/5 | Original | 66.20% | 67.76% | 67.00% | 70.22% |
| | ERwP | 0.00% | 67.55% | 0.00% | 69.63% |

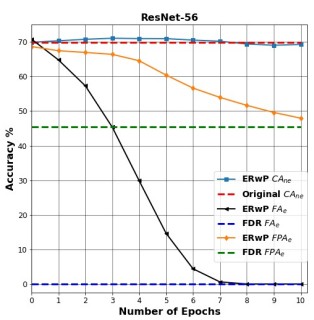

Figure 5: Plot denoting the performance of our proposed ERwP during the optimization process.

classes constitute the non-excluded classes. In order to show the effectiveness of our approach, we sort the absolute gradients of the parameters in the model (obtained through backpropagation for the excluded class augmented images) and choose a set of high relevance and low relevance parameters. We then prune/zero out these parameters and record the forgetting accuracy. Fig. 4 demonstrates that as we zero out the high relevance parameters, the forgetting accuracy of the excluded class drops by a huge margin. It also shows that as we zero out the low relevance parameters, there is only a minor change in the forgetting accuracy of the excluded class. This validates our approach for identifying the high relevant parameters for the restricted classes.

## 9.2 SIGNIFICANCE OF THE COMPONENTS OF THE PROPOSED ERwP APPROACH

We perform ablations on the CIFAR-100 dataset using the ResNet-56 model to study the significance of the $\mathcal{L}_c^e$, $\mathcal{L}_c^{ne}$ and $\mathcal{L}_{kd}$ components of our proposed ERwP approach. Table 3 indicates that optimizing the restricted class relevant parameters using only $\mathcal{L}_c^{ne}$ cannot significantly remove the information regarding the restricted classes from the model. Applying $\mathcal{L}_c^{ne}$ along with $\mathcal{L}_c^e$ significantly reduces the forgetting accuracy FA$_e$ and forgetting prototype accuracy FPA$_e$ but also significantly reduces the constraint accuracy CA$_{ne}$. Finally, applying the $\mathcal{L}_{kd}$ loss along with $\mathcal{L}_c^{ne}$ and $\mathcal{L}_c^e$ significantly reduces FA$_e$ and FPA$_e$ while maintaining the constraint accuracy CA$_{ne}$ very close to that of the original model.

## 9.3 ABLATION ON THE NUMBER OF EXCLUDED CLASSES

We report the experimental results for our approach for different splits of excluded and remaining classes of the CIFAR-100 dataset in Table 5. We observe that our ERwP performs well for all the splits for both the ResNet-20 and ResNet-56 architectures.

## 9.4 PERFORMANCE OF ERwP OVER TRAINING EPOCHS

We analyze the change in the performance of the model after every epoch of our proposed ERwP approach in Fig. 5. We observe that as the training progresses the constraint accuracy is maintained close to that of the original model, the forgetting accuracy keeps dropping till it reaches 0% and the forgetting prototype accuracy keeps falling and comes closer to that of the FDR model.

## 10 CONCLUSION

In this paper, we present a "Restricted Category Removal from Model Representations with Limited Data" problem in which the objective is to remove the information regarding a set of excluded/restricted classes from a trained deep learning model without hurting its predictive power for the remaining classes. We propose several baseline approaches and also the performance metrics for this setting. First, we propose a novel approach to identify the model parameters that are highly relevant to the restricted classes. Next, we propose a novel efficient approach that optimizes these model parameters in order to remove the restricted class information and re-use these parameters for the remaining classes. We experimentally show how our approach significantly outperforms all the proposed baselines and performs similar to the full data retraining model.

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

## 11 APPENDIX

### 11.1 PROCESS FOR SELECTING THE RESTRICTED CLASS RELEVANT PARAMETERS

First, we apply a data augmentation technique $f$, not used during training, to the images of the given restricted class. Next, we combine the predictions for these images and perform backpropagation. Finally, we select the parameters with the highest absolute gradient as the relevant parameters for the corresponding restricted class. Specifically, for a given restricted class, we choose the minimum number of such parameters from each network layer such that pruning these parameters will result in the maximum degradation of model performance on that restricted class. We use a process similar to the binary search for automatically selecting the parameters with the highest absolute gradient. We use an automated script that first creates a list of parameters in each layer, sorts them in descending order according to the gradient values, and checks if zeroing out the weights of the first 5% parameters from this list leads to near zero accuracy for that class. If not, then we select double the number of parameters chosen earlier and repeat this process. If the accuracy is near zero, we repeat the process with half the number of parameters chosen earlier. Please note, this process is just for identifying the parameters relevant to the restricted classes, and their weights are restored after this process. The combined set of the relevant parameters for all the excluded classes is referred to as the restricted/excluded class relevant parameters.

### 11.2 BASELINES

We propose several baseline models for the RCRMR-LD problem and compare our proposed approach with them. The baseline approaches are defined as follows:

**Original model:** It refers to the original model that is trained on the complete training set containing all the training examples from both the excluded and non-excluded classes. It represents the model that has not been modified by any technique to remove the excluded class information.

**Baseline 1 - Weight Deletion (WD):** It refers to the original model with a modified fully-connected classification layer. Specifically, we remove the weights corresponding to the excluded classes in the fully-connected classification layer so that it cannot classify the excluded classes.

**Baseline 2 - Training from Scratch on Limited Non-Restricted Class data (TSLNRC):** In this baseline, we train a new model from scratch using the limited training examples of only the non-excluded classes. It uses the complete training schedule as the original model and only uses the classification loss for training the model.

**Baseline 3 - Training from Scratch on Limited Non-Restricted Class data with KD (TSLNRC-KD):** This baseline is the same as baseline 2, but in addition to the classification loss, it also uses a knowledge distillation loss to ensure that the non-excluded class logits of the model (student) match that of the original model (teacher).

**Baseline 4 - Training of Original model on Limited Non-Restricted Class data (TOLNRC):** This baseline is the same as baseline 2, but the model is initialized with the weights of the original model instead of randomly initializing it.

**Baseline 5 - Training of Original model on Limited Non-Restricted Class data with KD (TOLNRC-KD):** This baseline is the same as baseline 4, but in addition to the classification loss, it also uses a knowledge distillation loss.

**Baseline 6 - Fine-tuning of Original model on Limited data after Mapping Restricted Classes to a Single Class (FOLMRCSC):** In this baseline approach, we first replace all the excluded class labels in the limited training data with a new single excluded class label and then fine-tune the original model for a few epochs on the limited training data of both the excluded and remaining classes. In the case of the examples from the excluded classes, the model is trained to predict the new single excluded class. In the case of the examples from the remaining classes, the model is trained to predict the corresponding non-excluded classes.

**Baseline 7 - Fine-tuning of Original model on Limited data after Mapping Restricted Classes to a Single Class with KD (FOLMRCSC-KD):** This baseline is the same as baseline 6, but in

addition to the classification loss, it also uses a knowledge distillation loss to ensure that the non-excluded class logits of the model (student) match that of the original model (teacher).

**Baseline 8 - Fine-tuning of Original model on Limited Non-Restricted Class data (FOLNRC):** In this baseline approach, we fine-tune the original model for a few epochs on the limited training data of non-excluded/remaining classes. The model is trained to predict the corresponding excluded classes of the training examples.

**Baseline 9 - Fine-tuning of Original model on Limited Non-Restricted Class data with KD (FOLNRC-KD):** This baseline is the same as baseline 8, but in addition to the classification loss, it also uses a knowledge distillation loss.

### 11.3 DATASETS

For the RCRMR-LD problem setting, we modify the CIFAR-100 (Krizhevsky et al., 2009), CUB (Wah et al., 2011) and ImageNet-1k (Russakovsky et al., 2015) datasets. In order to simulate the RCRMR-LD problem setting with limited training data, we choose the last 20 classes of the CIFAR-100 dataset as the excluded classes and take only 10% of the training images of each class. Similarly, we choose the last 20 classes of the CUB dataset as the excluded classes with only 3 training images per class. For ImageNet-1K, we choose the last 100 classes as the excluded classes with 5% of the training images to simulate the limited data available for this problem setting.

### 11.4 IMPLEMENTATION DETAILS

In this section, we provide all the details required to reproduce our experimental results. We use the ResNet-20 (He et al., 2016), ResNet-56, ResNet-164 architectures for the experiments on the CIFAR-100 dataset. We use the standard data augmentation methods of random cropping to a size of $32 \times 32$ (zero-padded on each side with four pixels before taking a random crop) and random horizontal flipping, which is a standard practice for training a model on CIFAR-100. In order to obtain the original and FDR models for the CIFAR-100 dataset, we train the network for 300 epochs with a mini-batch size of 128 using the stochastic gradient descent optimizer with momentum 0.9 and weight decay $1e - 4$. We choose the initial learning rate as 0.1, and we decrease it by a factor of 5 after every 50 epochs. For the CIFAR-100 experiments with ERwP using the ResNet-20, ResNet-56, and ResNet-164 architectures, we take learning rate$= 1e - 4$, $\beta = 10$ and optimize the network for 10 epochs. Since the available limited training data is only 10% of the entire CIFAR-100 dataset, therefore, our ERwP approach is approximately $30 * 10 = 300\times$ faster than the FDR method.

For the experiments on the ImageNet dataset, we use the ResNet-18, ResNet-50, and MobileNet-V2 architectures. We use the standard data augmentation methods of random cropping to a size of $224 \times 224$ and random horizontal flipping, which is a standard practice for training a model on ImageNet-1k. In order to obtain the original and FDR models for the ImageNet dataset, we train the network for 100 epochs with a mini-batch size of 256 using the stochastic gradient descent optimizer with momentum 0.9 and weight decay $1e - 4$. We choose the initial learning rate as 0.1, and we decrease it by a factor of 10 after every 30 epochs. For evaluation, the validation images are subjected to center cropping of size $224 \times 224$. For the ImageNet-1k experiments (5% training data) with ERwP using the ResNet-50 architecture, we optimize the network for 10 epochs with a learning rate of $9e - 5$ using $\beta = 200$. For the ERwP experiments using the ResNet-18 architecture, we optimize the network for 10 epochs using $\beta = 200$ with an initial learning rate of $1.1e - 4$ and a learning rate of $1.1e - 5$ from the third epoch onward. In the case of the ERwP experiments with the MobileNet-V2 architecture, we optimize the network for 10 epochs using $\beta = 400$ with an initial learning rate of $1.5e - 4$ and a learning rate of $1.5e - 5$ from the third epoch onward. Since the available limited training data is only 5% of the entire ImageNet-1k dataset, therefore, our ERwP approach is approximately $20 * 10 = 200\times$ faster than the FDR method. For the experiments on the CUB-200 dataset, we use the ResNet-50 architecture pre-trained on the ImageNet dataset. In order to obtain the original and FDR models for the CUB-200 dataset, we train the network for 50 epochs with a mini-batch size of 64 using the stochastic gradient descent optimizer with momentum 0.9 and weight decay $1e - 3$. We choose the initial learning rate as $1e - 2$, and we decrease it by a factor of 10 after epochs 30 and 40. For the CUB-200 experiments (3 images per class, i.e., 10% training data) with ERwP using the ResNet-50 architecture, we optimize the network for 10 epochs with a learning rate of $1e - 4$ using $\beta = 10$. Since the available limited training data is only 10% of the

Table 6: Experimental results on the CUB dataset with ResNet-50 architecture for the RCRMR-LD problem with 20 excluded classes using only 3 training images per class.

| Methods | $FA_e$ | $FPA_e$ | $CA_{ne}$ |
|---|---|---|---|
| Original | 85.20% | 84.69% | 77.37% |
| **No Training** | | | |
| Baseline 1 - WD | 0.00% | 85.71% | 77.64% |
| **Full Train Schedule** | | | |
| Baseline 2 - TSLNRC | 0.00% | 30.27% | 27.56% |
| Baseline 3 - TSLNRC-KD | 0.00% | 35.54% | 31.66% |
| Baseline 4 - TOLNRC | 0.00% | 60.37% | 64.60% |
| Baseline 5 - TOLNRC-KD | 0.00% | 68.37% | 70.48% |
| **Only Fine-Tuning** | | | |
| Baseline 6 - FOLMRCSC | 53.40% | 77.38% | 74.39% |
| Baseline 7 - FOLMRCSC-KD | 60.88% | 81.12% | 75.14% |
| Baseline 8 - FOLNRC | 84.86% | 84.18% | 76.85% |
| Baseline 9 - FOLNRC-KD | 84.35% | 85.20% | 77.70% |
| **ERwP (Ours)** | 0.77% | 48.89% | 75.45% |

entire CUB-200 dataset, therefore, our ERwP approach is approximately $5 * 10 = 50\times$ faster than the FDR method.

In our proposed approach, we use $\kappa = 2$ for all the experiments. We use a popular Pytorch implementation[1] for performing knowledge distillation. We run all the experiments 3 times (using different random seeds) and report the average accuracy. We perform all the experiments using the Pytorch framework version 1.6.0 (Paszke et al., 2017) and Python 3.0. We use 4 GeForce GTX 1080 Ti graphics processing units for our experiments.

## 11.5 RCRMR-LD PROBLEM IN CUB-200 CLASSIFICATION

Table 6 reports the experimental results for different approaches to the RCRMR-LD problem over the CUB dataset using the ResNet-50 architecture. Our proposed ERwP approach achieves a constraint accuracy $CA_{ne}$ that is very close to that of the original model even though we use only 3 images per class for optimizing the model. It achieves close to 0% forgetting accuracy $FA_e$ and achieves a $FPA_e$ that is significantly lower than that of the original model by an absolute margin of 35.80%. Similar to the CIFAR-100 experiments, our ERwP approach outperforms all the baseline approaches. Further, our ERwP requires only 10 epochs to remove the excluded class information from the model. Since the available limited training data is only 10% of the entire CUB dataset, therefore, our ERwP approach is approximately $5 * 10 = 50\times$ faster than the FDR method that is trained on the full training data for 50 epochs.

## 11.6 ABLATION EXPERIMENTS FOR $\beta$ AND $\kappa$

We perform ablation experiments to identify the most suitable values for the hyper-parameters $\beta$ and $\kappa$ for our proposed ERwP. The ablation results in Tables 7, 8, validate our choice of hyper-parameter values considering the forgetting accuracy and the constraint accuracy of the resulting model.

## 11.7 EFFECT OF DIFFERENT DATA AUGMENTATIONS ON THE IDENTIFICATION OF CLASS RELEVANT MODEL PARAMETERS

We perform experiments to verify our approach of identifying the highly relevant parameters for any restricted class using various augmentation techniques (grayscale, vertical flip, rotation, random affine augmentations). We chose the same restricted class of CIFAR-100 and used the ResNet-56 network for all the experiments. The results in Fig. 6 indicate that for all the compared data augmentations approaches, pruning/zeroing out the high relevance parameters obtained using our approach, results

---

[1]https://github.com/peterliht/knowledge-distillation-pytorch/blob/master/model/net.py

Table 7: Experimental results on the CIFAR-100 dataset with ResNet-20 architecture for the RCRMR-LD problem with 20 excluded classes using our proposed ERwP with different values of $\beta$.

| $\beta$ | Methods | $FA_e$ | $CA_{ne}$ |
|---|---|---|---|
| - | Original | 70.15% | 67.06% |
| 8 | ERwP | 0.00% | 66.03% |
| 9 | ERwP | 0.00% | 66.23% |
| 10 | ERwP | 0.00% | **66.79**% |
| 11 | ERwP | 0.00% | 66.58% |
| 12 | ERwP | 0.00% | 66.15% |

Table 8: Experimental results on the CIFAR-100 dataset with ResNet-20 architecture for the RCRMR-LD problem with 20 excluded classes using our proposed ERwP with different values of $\kappa$.

| $\kappa$ | Methods | $FA_e$ | $CA_{ne}$ |
|---|---|---|---|
| - | Original | 70.15% | 67.06% |
| 1.0 | ERwP | 0.00% | 66.05% |
| 1.5 | ERwP | 0.00% | 66.08% |
| 2.0 | ERwP | 0.00% | **66.79**% |
| 2.5 | ERwP | 0.00% | 66.30% |
| 3.0 | ERwP | 0.00% | 66.23% |

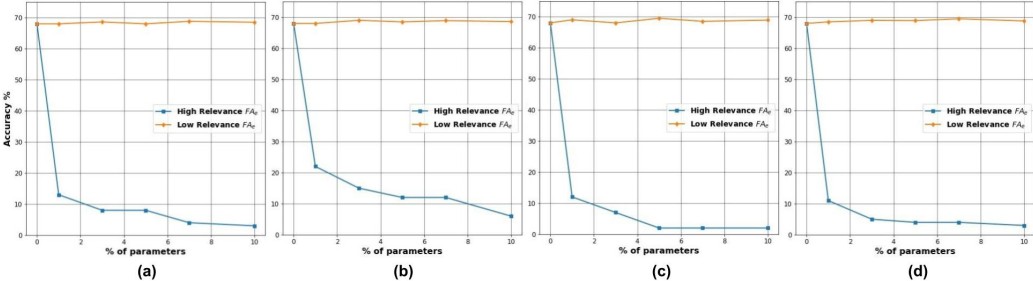

Figure 6: Ablation to validate our approach for identifying restricted class relevant model parameters using different augmentation techniques w.r.t. the same randomly chosen restricted class of CIFAR-100. We use the ResNet-56 network for these experiments. The data augmentation techniques used are (a) grayscale augmentation, (b) vertical flip augmentation, (c) rotation augmentation, (d) random affine augmentation. In each case, the figure shows the model performance when the low relevance and high relevance parameters obtained using our approach are zeroed out.

in a huge drop in the forgetting accuracy of the excluded class. Further, zeroing out the low relevance parameters, has a minor impact on the forgetting accuracy of the excluded class.

## 11.8 ABLATION EXPERIMENTS ON THE RESTRICTED CLASS RELEVANT PARAMETERS

We perform ablation experiments with ERwP to check if only 25% and 50% of the restricted class relevant parameters of each layer identified using our proposed procedure can be used for ERwP. We run each of these experiments for the same number of epochs for the CIFAR-100 dataset and ResNet-56 network. However, we observed that the final $FPA_e$ falls from 68.65% to 60.35% and 53.7%, respectively, for 25% and 50% of restricted class relevant parameters of each layer as compared to 47.84% when using all the restricted class relevant parameters per layer identified using our approach. The good performance of our approach is more evident in light of the performance of the FDR model that achieves a $FPA_e$ accuracy of 45.40%. We provide this result as a reference to demonstrate that the 47.84% $FPA_e$ accuracy is due to the generalization power of the model and not due to the restricted classes information in the model. This shows that our approach effectively identifies the class-relevant parameters of the model for a given class.

## 11.9 PERFORMANCE OF ERwP OVER TRAINING EPOCHS

We analyze the change in the performance of the model after every epoch of our proposed ERwP approach in Fig. 7 for the CIFAR-100 dataset with 20 excluded classes using the ResNet-20 and ResNet-56 architectures. For both architectures, we observe that as the training progresses, ERwP maintains the constraint accuracy close to that of the original model and forces the forgetting accuracy to drop to $0\%$. ERwP also forces the forgetting prototype accuracy to keep dropping and makes it similar to the FDR model.

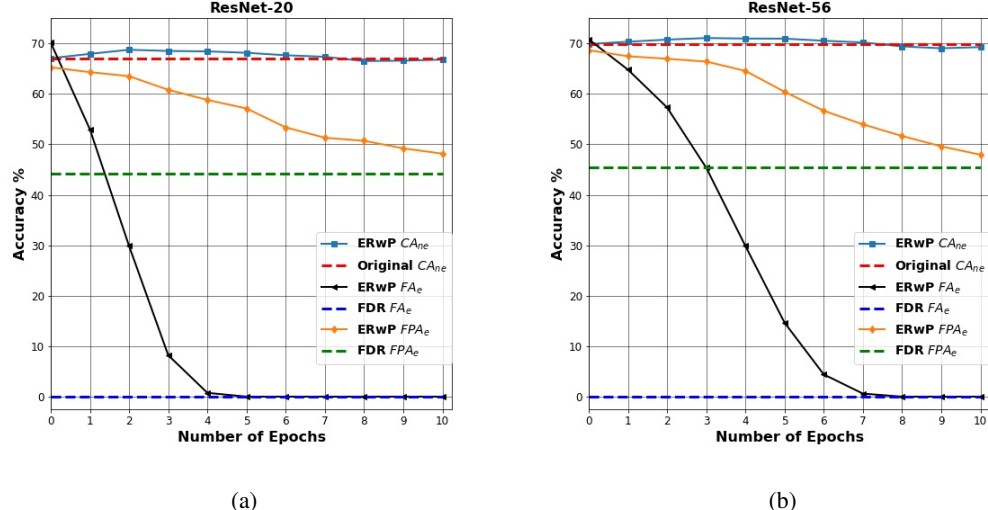

(a)                                                                              (b)

Figure 7: Plots denoting the performance of our proposed ERwP during the optimization process for forgetting 20 excluded classes from CIFAR-100 using a) ResNet-20 and b) ResNet-56 architectures.

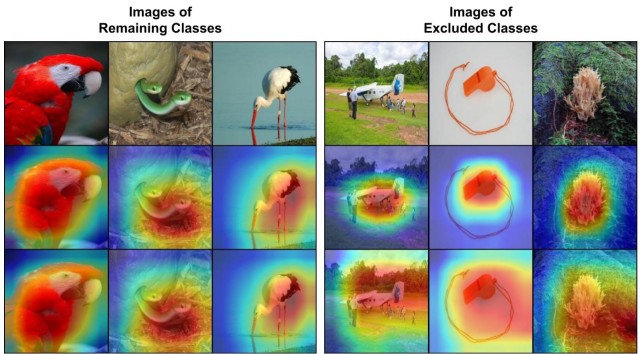

Figure 8: Class activation maps of ImageNet images from the excluded and non-excluded classes, for the original ResNet-50 (second row) and ResNet-50 after applying our proposed ERwP approach (third row). First row depicts the real images.

## 11.10 FINETUNING RESTRICTED CLASS RELEVANT PARAMETERS ON REMAINING CLASSES

In our experimental results, we demonstrated how finetuning the model on the limited training data of the non-excluded classes cannot sufficiently remove the excluded class information from the model. We also perform an ablation experiment to demonstrate that finetuning only the restricted class relevant parameters using the limited training data of the non-excluded classes is also not effective in sufficiently removing the excluded class information from the model. We perform these experiments on the CIFAR-100 dataset with the ResNet-56 architecture. We observe the constraint accuracy $CA_{ne}$, forgetting accuracy $FA_e$ and the forgetting prototype accuracy $FA_e$ are almost the same even in this case. Therefore, finetuning only on the remaining class data cannot sufficiently remove the excluded class information from the model representations.

## 11.11 EFFECT OF USING THE PROPOSED ERWP APPROACH WHEN THE ENTIRE DATASET IS AVAILABLE

We perform ablation experiments to demonstrate the performance of our proposed ERwP approach when the entire training data is available. We perform these experiments on the CIFAR-100 dataset using ResNet-20 and ResNet-56. We observe experimentally that for both the ResNet-20 and ResNet-

56 experiments using ERwP, the forgetting accuracy $\texttt{FA}_\texttt{e}$ accuracy is 0% and the constraint accuracy $\texttt{CA}_\texttt{ne}$ matches that of the original model. Further, the gap between the forgetting prototype accuracy $\texttt{FPA}_\texttt{e}$ of ERwP and the FDR model reduces from 3.86% (for limited data) to 2.79% for ResNet-20. Similarly, the gap reduces from 2.44% (for limited data) to 1.65% for ResNet-56. However, ERwP requires only 2-3 epochs of optimization ($\sim$100-150$\times$ faster than the FDR model) for achieving this performance when trained on the entire dataset. This makes it significantly faster than any approach that trains on the entire dataset.

### 11.12 QUALITATIVE ANALYSIS

In order to analyze the effect of removing the excluded class information from the model using our proposed ERwP approach, we study the class activation map visualizations (Selvaraju et al., 2017) of the model before and after applying ERwP. We observe in Fig. 8 that for the images from the excluded classes, the model's region of attention gets scattered after applying ERwP, unlike the images from the remaining classes.

