# OpenReview forum: "Restricted Category Removal from Model Representations using Limited Data"
_ICLR.cc/2022/Conference — ICLR 2022 Submitted_

### Official Review · Reviewer_Y25C · 2021-10-31

**Correctness:** 3
**Technical Novelty And Significance:** 2
**Empirical Novelty And Significance:** 4
**Recommendation:** 6
**Confidence:** 4

**Main Review:**

Pros:

1. The problem RCRMR-LD seems interesting and practical, which addresses the specific class-level restriction by removing corresponding model representations. This setting also save time and computational resource for large scale datasets.

2. Experiments in this paper are solid and convincing enough. They design 5 basic baselines and perform corresponding ablation study. Considering RCRMR-LD is a new problem, if there exists, comparing some related works will be better.

################################################

Cons:

1. From my point of view, the transformation $f$ plays a key role in identifying the parameters that are highly relevant to the
restricted classes. However, they seem only try the grayscale transformation and do not give more discussion about $f$. If the model is just trained by grayscale images, will this method fail? For natural language tasks, what transformation are you going to use? I suggest that the authors make more discussion and comparison of the various transformations.

2. From Table 1, I find all the FPA$_e$ of ERwP are relatively high, indicating that the feature representations of the model still contain much restricted category information. Although they indeed remove restricted category from class-level, attackers still can use some model inversion techniques (e.g., [1]) to restore the restricted class data with few owned ones, leading to privacy leakage.

3. Identifying those parameters that are relevant to the restricted classes through ERwP is still heuristic. I admit that ERwP seems to make sense, but some verifications about this claim need to be included.

4. Except for "Related work", I do not find any references in this paper. At least in the "Introduction", you should cite some related works to support your claims.

5. Colloquial expressions and grammar issues are common, thus the writting needs further improvement.

#########################################################

Typo:
1. "Baseline 4 - Training of Original model on Limited Non-Restricted Class data with (TOLNRC):" -- missing words?

2. "$N_e$ and $N_r$ refer to the number of excluded classes, respectively." -- missing words?

 and so on...

############################################################

Questions during rebuttal period:

Please address and clarify the cons above.

##############################################################

References:

[1] Zhang Y, Jia R, Pei H, et al. The secret revealer: Generative model-inversion attacks against deep neural networks. CVPR, 2020.

**Summary Of The Paper:**

This paper proposes a novel and practical problem called RCRMR-LD, aiming to removel restricted categories from model representations with limited data. They first give some direct solutions and analyze their weaknesses. Then, they propose their own solution to discard the restricted class information from the restricted class relevant parameters. Experiments verify that this approach not only performs similar to FDR but also is faster than it.

**Summary Of The Review:**

The setting proposed by this paper is novel and practical. However, there exists some technical flaws that need to bu further solved.

Please see the "Main Review" for details.

---

> ### Author Response · Authors · 2021-11-23
> **Response to Reviewer 4/ Y25C**
>
> Thanks for your positive comments about our proposed problem setting, extensive evaluation and the effectiveness of our approach in removing the restricted class information from the model parameters. We have responded to your concerns below. We sincerely hope our response will help you rethink your assessment. We have thoroughly updated the paper to address your concerns. Please update your review accordingly.
>
> R4.1. From my point of view, the transformation f plays a key role in identifying the parameters that are highly relevant to the restricted classes. However, they seem only try the grayscale transformation and do not give more discussion about f. If the model is just trained by grayscale images, will this method fail? For natural language tasks, what transformation are you going to use? I suggest that the authors make more discussion and comparison of the various transformations.
>
>
> Ans: We apologize for this confusion. We have provided a detailed discussion in point C2 of the common concerns regarding the effect of different data augmentations on the selection process of the restricted class relevant parameters.

---

> > ### Author Response · Authors · 2021-11-23
> > **Response to Reviewer 4/Y25C (contd.)**
> >
> > R4.2. From Table 1, I find all the $\texttt{FPA}_\texttt{e}$ of ERwP are relatively high, indicating that the feature representations of the model still contain much restricted category information. Although they indeed remove restricted category from class-level, attackers still can use some model inversion techniques (e.g., [1]) to restore the restricted class data with few owned ones, leading to privacy leakage.
> >
> > Ans: The $\texttt{FPA}_\texttt{e}$ accuracy obtained using ERwP is significantly lower than the original model, e.g., $\texttt{FPA}_\texttt{e}$ of ERwP is 47.84\% compared to 68.65\% of the original model for the CIFAR-100 dataset using the ResNet-56 model. However, this does not indicate the presence of much restricted category information. This is because the $\texttt{FPA}_\texttt{e}$ accuracy involves creating prototypes from the limited training data of the restricted classes and the remaining classes and finding the nearest neighbor class. Therefore, this process is dependent on the features generated by the deep learning model. Deep learning models generally produce highly discriminative features that can be used to create good prototype classifiers even for classes that the models were not trained on. For example, in the few-shot learning setting, the model is generally trained only on the base classes and then evaluated on novel class episodes using a prototype-based classifier. The prototype-based classifier of the few-shot learning setting is very effective in classifying the novel classes even though the deep model, which was used to obtain the features for the prototypes, was never trained on the novel classes. The discriminative nature of the features produced by deep learning models is the main reason why ImageNet pre-trained model features are used to train classifiers for other datasets and settings, such as in zero-shot learning. In order to better appreciate the effectiveness of our approach, we also consider the FDR model, which has not seen any training data of the restricted classes and still achieves a $\texttt{FPA}_\texttt{e}$ accuracy close to that of our approach, e.g. FDR achieves a $\texttt{FPA}_\texttt{e}$ accuracy of 45.40\% for the CIFAR-100 dataset using the ResNet-56 model while our approach achieves an $\texttt{FPA}_\texttt{e}$ accuracy of 47.84\%. We provide this result as a reference to demonstrate that the non-zero $\texttt{FPA}_\texttt{e}$ accuracy of ERwP is due to the generalization power of the model and not due to the restricted classes information in the model.
> >
> >
> >
> > The generative model inversion attack [1], trains a GAN to learn to generate images that have a high likelihood under the model [1]. Since our proposed ERwP method has already modified the model such that the $FA_e$ accuracy is close to zero, the fully-connected classification layer cannot classify the restricted classes anymore. Therefore, the generative model inversion process will not be able to use the classification layer to learn to generate the images from the restricted classes. Since our approach has also removed the restricted class information from the model parameters, the prototype-based classifier is also not effective, and the $\texttt{FPA}_\texttt{e}$ accuracy of the model is due to the generalization power of the model. The prototype-based classifier based on our model will behave as if the model was never trained on the restricted classes similar to FDR. Therefore, the model obtained after applying our approach, is not effective for generative model inversion, since, the generative model inversion method requires a model with high predictive power (as demonstrated in [1]) for the target classes even though the attacker has a few images from those classes.
> >
> > We have provided a detailed response above in order to address your concern. However, addressing a generative model inversion attack is not an objective of our problem setting.
> >
> > References:
> > [1] Zhang Y, Jia R, Pei H, et al. The secret revealer: Generative model-inversion attacks against deep neural networks. CVPR, 2020.

---

> > > ### Author Response · Authors · 2021-11-23
> > > **Response to Reviewer 4/Y25C (contd.)**
> > >
> > > R4.3. Identifying those parameters that are relevant to the restricted classes through ERwP is still heuristic. I admit that ERwP seems to make sense, but some verifications about this claim need to be included.
> > >
> > > Ans: We have shown in Fig. 4 of the paper that choosing the highly relevant parameters for a class using our approach and zeroing them out leads to a drastic fall in the model performance of that class. This verifies our claim that our approach identifies the relevant parameters for a given class. Additionally, we have now shown that selecting less number of parameters for ERwP from the restricted class relevant parameters identified using our approach results in a lower reduction in the forgetting accuracy of the restricted classes (Refer to our response to point R3.5. of reviewer 3/pyXW). We have also shown in Fig. 6 in the appendix of the revised paper, the effect of using different types of augmentation on the identification of restricted class relevant parameters. We have provided a detailed discussion in point C2 of the common concerns to clarify the doubts regarding the use of different data augmentations/transformations in the selection process of the restricted class relevant parameters.
> > >
> > >
> > > R4.4. Except for "Related work", I do not find any references in this paper. At least in the "Introduction", you should cite some related works to support your claims. Colloquial expressions and grammar issues are common, thus the writing needs further improvement.
> > >
> > > Ans: Thank you for your suggestions. We have thoroughly revised our paper to incorporate all your suggestions.

---

> > > > ### Comment · Reviewer_Y25C · 2021-11-27
> > > > **Thank you for your clarification**
> > > >
> > > > I have carefully read the authors' response and other reviewers' comments, and my concerns have been addressed well. Anyhow, this paper proposes an interesting and practical setting, and I think this setting will be further studied in the future. I am willing to increase my score to 6.

---

> > > > > ### Author Response · Authors · 2021-11-29
> > > > > **Thank you for your response**
> > > > >
> > > > > Dear Reviewer,
> > > > >
> > > > > We thank you for the time invested by you in carefully reading our revised paper and responses. Your comments helped us to improve our paper. We have addressed your concerns, and you have already mentioned that our problem setting is new and interesting. Further, you have also mentioned that our experimental results are solid and convincing. Will you kindly consider increasing your score since ICLR is a reputed conference that promotes new problem settings, and ideas. Please update the rating in your review accordingly.
> > > > >
> > > > > Regards,
> > > > >
> > > > > Authors

---

### Official Review · Reviewer_pyXW · 2021-10-31

**Correctness:** 3
**Technical Novelty And Significance:** 3
**Empirical Novelty And Significance:** 2
**Recommendation:** 6
**Confidence:** 4

**Main Review:**

Introduction:
The paper is well written, and the evaluation is very detailed. It is an interesting idea to remove class information from the model with a limited amount of data. However, from the description in the paper, it is not clear why this is a real-world problem. It would be beneficial to rate the importance of this application if the authors had provided sources for such cases or a more detailed description of a specific scenario.

Method:
The re-training procedure of the model with only a limited amount of training data is very detailed. The description of the identification of the relevant parameters for the restricted classes is missing some details. For example, it is not defined what other transformation besides the grayscale transformation is used. If other transformations are optional, it would be good to know what type of transformations are used in the experiments. Furthermore, it is not clear how the parameters with the highest gradients are selected. Is a fixed threshold used? What is the minimum number of parameters of each layer that are selected? Is this a fixed number for each class? Does it depend on the number of excluded classes?

Evaluation:
The evaluation is very detailed, with eight baseline methods to show the performance of the presented method. However, the results of the FDR model are not shown in Table 1, only mentioned in the text. Is there a reason for this? Adding the results of the FDR to Table 1 would be beneficial. It would be very interesting to see how the parameter selection influences the accuracy of the model. Unfortunately this is not part of the evaluation.


**Summary Of The Paper:**

In this paper, the authors present a new method to remove information about specific classes from a trained model without reducing the performance of the remaining classes. After the information is removed, the model should not be able to identify the class anymore. Instead of retraining the complete model from scratch without the restricted classes, the presented method only needs a few examples of the restricted classes and the remaining classes. In terms of speed, the presented method is ~200 times faster on ImageNet than a new model training without the restricted classes. Furthermore, they present a method for identifying model parameters that are mainly relevant to the restricted classes. The evaluation of the model is performed on the CIFAR-100, ImageNet-1k, and the CUB-200 dataset. For a detailed comparison, eight baseline methods were designed and evaluated. An ablation study is performed on the class relevant parameters and the number of classes that are excluded. The presented method achieves an accuracy close to the original model on the remaining classes in terms of accuracy. Also, the forgetting prototype accuracy is close to the model trained only on the remaining classes.


**Summary Of The Review:**

The presented method of re-training a model to forget a specific class is very interesting. However, the part for identifying the most relevant model parameter is missing some essential details. For example, how the parameters are selected (manually or automatic) or the number of selected parameters. This information is essential to understand the method. Moreover, the influence of the parameter selection method is not studied in the evaluation part.

---

> ### Author Response · Authors · 2021-11-23
> **Response to Reviewer 3/ pyXW**
>
> Thanks for your positive comments about our proposed problem setting, extensive evaluation and the effectiveness of our approach in removing the restricted class information from the model parameters. We have responded to your concerns below. We sincerely hope our response will help you rethink your assessment. We have thoroughly updated the paper to address your concerns. Please update your review accordingly.
>
> R3.1. However, from the description in the paper, it is not clear why this is a real-world problem. It would be beneficial to rate the importance of this application if the authors had provided sources for such cases or a more detailed description of a specific scenario.
>
> Ans: We have provided a detailed discussion in point C1 of the common concerns regarding the practical real-world scenarios which can be affected by our RCRMR-LD problem.
>
> R3.2. The description of the identification of the relevant parameters for the restricted classes is missing some details. For example, it is not defined what other transformation besides the grayscale transformation is used. If other transformations are optional, it would be good to know what type of transformations are used in the experiments
>
> Ans: We apologize for this confusion. We have provided a detailed discussion in point C2 of the common concerns regarding the effect of different data augmentations on the selection process of the restricted class relevant parameters. We have included this discussion in Sec. 11.7 in the appendix of the revised paper.
>
> R3.3. Furthermore, it is not clear how the parameters with the highest gradients are selected. Is a fixed threshold used? What is the minimum number of parameters of each layer that are selected? Is this a fixed number for each class? Does it depend on the number of excluded classes?
>
> Ans: We apologize for this confusion. There are no fixed thresholds for selecting the parameters with the highest gradients. Instead, we use a process similar to the binary search for automatically selecting the parameters. We use an automated script that first creates a list of parameters in each layer, sorts them in descending order according to the gradient values, and checks if zeroing out the weights of the first 5\% parameters from this list leads to near zero accuracy for that class. If not, then we select double the number of parameters chosen earlier and repeat this process. If the accuracy is near zero, we repeat the process with half the number of parameters chosen earlier. Please note, this process is just for identifying the parameters relevant to the restricted classes, and their weights are restored after this process. We have now added these details in the Sec. 11.1 in the appendix of the revised paper for better understanding. There is no minimum number of parameters that are chosen for any layer for a class since the parameters are automatically identified using the binary search procedure mentioned above. Further, since the parameters are learned in a shared manner, many parameters will be shared by multiple classes. So we cannot definitively specify how the number of restricted class-relevant parameters depends on the number of excluded classes. With an increase in the number of excluded classes, the possibility of common parameters being selected becomes even higher. Therefore, as the number of excluded classes increases, the number of restricted class relevant parameters will also increase, but due to the sharing of parameters among classes, the exact increase cannot be predicted. Please also refer to our response to your concern R3.5.
>
> R3.4. The evaluation is very detailed, with eight baseline methods to show the performance of the presented method. However, the results of the FDR model are not shown in Table 1, only mentioned in the text. Is there a reason for this? Adding the results of the FDR to Table 1 would be beneficial.
>
> Ans: Please refer to our response to R2.5 of Reviewer 2/3V1Q.

---

> > ### Author Response · Authors · 2021-11-23
> > **Response to Reviewer 3/ pyXW  (contd.)**
> >
> > R3.5. It would be very interesting to see how the parameter selection influences the accuracy of the model. Unfortunately this is not part of the evaluation.
> >
> > Ans: The parameter selection process is not manual but automated. The process approximately selects the minimum number of high relevance parameters of restricted class in a layer that on zeroing leads to the model accuracy for a restricted class to become near zero. We have also performed experiments with ERwP while using only 25\% and 50\% of the restricted class relevant parameters of each layer identified using our proposed procedure. We ran each of these experiments for the same number of epochs for the CIFAR-100 dataset and ResNet-56 network. However, we observed that the final $\texttt{FPA}_\texttt{e}$ falls from 68.65\% to 60.35\% and 53.7\%, respectively, for 25\% and 50\% of restricted class relevant parameters of each layer as compared to 47.84\% when using all the restricted class relevant parameters per layer identified using our approach. The good performance of our approach is more evident in light of the performance of the FDR model that achieves a $\texttt{FPA}_\texttt{e}$ accuracy of 45.40\%. We provide this result as a reference to demonstrate that the 47.84\% $\texttt{FPA}_\texttt{e}$ accuracy is due to the generalization power of the model and not necessarily due to the restricted classes information in the model. This shows that our approach effectively identifies the class-relevant parameters of the model for a given class. We have added these results in Sec. 11.8 of the appendix in the revised paper. Please refer to our response to the point R4.2 of reviewer 4/Y25C for a detailed discussion on the $\texttt{FPA}_\texttt{e}$ accuracy.

---

> > > ### Author Response · Authors · 2021-11-28
> > > **Follow-up**
> > >
> > > Dear Reviewer,
> > >
> > > We are eagerly waiting for your comments regarding our response to your review, so that we can properly address any further concerns before the deadline for discussion that is very close.
> > >
> > > Regards,
> > >
> > > Authors

---

> > > > ### Comment · Reviewer_pyXW · 2021-11-29
> > > > **Thank you for your response**
> > > >
> > > > I thank the authors for their response. After considering their response, I would increase my score to 6.

---

> > > > > ### Author Response · Authors · 2021-11-29
> > > > > **Thank you for your response to our comments**
> > > > >
> > > > > Dear Reviewer,
> > > > >
> > > > > We thank you for the time invested by you in carefully reading our revised paper and responses. Your comments helped us to improve our paper. We have addressed your concerns and you have already mentioned that our proposed method is new and very interesting. Further, you have also mentioned that our evaluation is very detailed and demonstrates the effectiveness of our method. Will you kindly consider increasing your score since ICLR is a reputed conference that promotes new problems settings and ideas. Please update the rating in your review accordingly.
> > > > >
> > > > > Regards,
> > > > >
> > > > > Authors

---

### Official Review · Reviewer_3V1Q · 2021-11-02

**Correctness:** 2
**Technical Novelty And Significance:** 2
**Empirical Novelty And Significance:** 3
**Recommendation:** 5
**Confidence:** 4

**Main Review:**

This paper proposes a new learning setting of fine-tuning a pretrained model to forget some specific categories which is motivated by class-level privacy. The solution to this challenge is firstly detecting the most related model parameters that significantly affect model performance on restricted classes and then tuning on a small number of examples with the losses of desired classification capability. The proposed method is experimentally demonstrated effective than possible baselines.

I mainly have the following concerns.
1. The motivation of the new setting is not strong. In introduction, the class-level privacy is specified by violated privacy concerns and corresponding examples. However, they are not quite convincing to me, and I feel more practical instances are needed to clarify the significance of studying class-level privacy. In particular, in what situation there would be only a few training examples available when considering removing information of restricted class from model concerned with privacy?

2. In related work, individual data deletion (Ginart et al 2019) is cited but not properly evaluated. Following the work of data deletion, I feel there also exists an important problem which is ignored. That is, making model forget some examples or some classes does not mean zero classification accuracy or random classification accuracy (i.e., 1/N). In data deletion work, Ginart et al tune the pretrained model by only compensating the impact of deleted samples instead of forcing model have large error on them. As a result, the tuned model turns out to be never seeing the deleted examples. This work obviously cannot guarantee it from the loss shown as Eq. 1. For example, a 3-way classifier on dog, cat, and leopard and leopard is the restricted class. It is predicted a classifier training on dog and cat only would intend classify leopard to cat because of their natural similarity. Thus, a careful clarification about this point is required in this paper, especially from the view of class-privacy.

3. The process of identifying parameters related to restricted classes seems quite empirically, as a transformation component is needed from some prior knowledge. The authors have mentioned it for images. However, many data privacy related data are also tabular. In this case, how to apply a proper transformation? If this component is quite related to data format, any workaround for this issue?

4. From Figure 3, KD is defined for remaining classes only, but the KD loss also includes restricted classes.

5. It is interesting to see the model performance comparison with the original training in term of remaining classes only (also related to concern 2,  the model performance on the original raining data of remaining classes only may be a good reference point for evaluation), although original training data might be inaccessible in the proposed setting.


**Summary Of The Paper:**

This paper proposes a new learning setting of fine-tuning a pretrained model to forget some specific categories which is motivated by class-level privacy. The solution to this challenge is firstly detecting the most related model parameters that significantly affect model performance on restricted classes and then tuning on a small number of examples with the losses of desired classification capability. The proposed method is experimentally demonstrated effective than possible baselines.

**Summary Of The Review:**

The paper has a weak motivation for the new setting. The proposed method seems too heuristic and the evaluation for the new setting is not appropriate.

---

> ### Author Response · Authors · 2021-11-23
> **Response to Reviewer 2 / 3V1Q**
>
> Thanks for your positive comments about the effectiveness of our approach in removing the restricted class information from the model parameters. We have responded to your concerns below. We sincerely hope our response will help you rethink your assessment.  We have thoroughly updated the paper to address your concerns. Please update your review accordingly.
>
>
> R2.1. The motivation of the new setting is not strong. In introduction, the class-level privacy is specified by violated privacy concerns and corresponding examples. However, they are not quite convincing to me, and I feel more practical instances are needed to clarify the significance of studying class-level privacy. In particular, in what situation there would be only a few training examples available when considering removing information of restricted class from model concerned with privacy?
>
> Ans: We have provided a detailed discussion in point C1 of the common concerns regarding the practical real-world scenarios where our problem setting can occur. We have included these examples in Secs. 1, 3 of the revised paper.
>
> R2.2. In related work, individual data deletion [1] is cited but not properly evaluated. Following the work of data deletion, I feel there also exists an important problem which is ignored. That is, making model forget some examples or some classes does not mean zero classification accuracy or random classification accuracy (i.e., 1/N). In data deletion work, Ginart et al tune the pretrained model by only compensating the impact of deleted samples instead of forcing model have large error on them. As a result, the tuned model turns out to be never seeing the deleted examples. This work obviously cannot guarantee it from the loss shown as Eq. 1. For example, a 3-way classifier on dog, cat, and leopard and leopard is the restricted class. It is predicted a classifier training on dog and cat only would intend classify leopard to cat because of their natural similarity. Thus, a careful clarification about this point is required in this paper, especially from the view of class-privacy.
>
> Ans: The individual data deletion [1] paper referred to by the reviewer deals with "data deletion in the context of a machine learning algorithm and model". It shows how to remove the influence of a data point from a k-means clustering model. Our work focuses on restricted category removal from deep learning models with limited data. Therefore, the approaches proposed in Ginart et al 2019 cannot be applied. Further, the objective of data deletion is to remove a data point without affecting the model performance on any classes, including the class of the deleted data point. This is in stark contrast to our RCRMR-LD problem, where the objective is to remove the knowledge of a set of classes or categories from the model. Further, data deletion methods will require access to the entire training data of a class in order to remove the entire knowledge of a class (refer to the appendix A.1. of [2]). This is because deep learning models have a high generalization power even on unseen examples of a class on which they have been trained, and simply deleting a few data points of a class from the knowledge base of the model will not be enough to forget that class. However, in our proposed problem setting, only a limited number of training examples are present for any class. Therefore, data-deletion approaches are not solutions to our proposed RCRMR-LD problem setting. This is why we have not evaluated the data deletion work.
>
> References:
>
> [1] Making ai forget you: Data deletion in machine learning, Ginart et al., 2019.
>
> [2] Mixed-privacy forgetting in deep networks, Golatkar et al. CVPR 2021.

---

> > ### Author Response · Authors · 2021-11-23
> > **Response to Reviewer 2 / 3V1Q  (contd.)**
> >
> > R2.2 (contd.)
> >
> > In eq 1 of our proposed approach, we define a loss $ L^e_{c}$ for optimizing the model on the training examples of the restricted classes using gradient ascent in order to remove the restricted class information from the model. However, only applying gradient ascent cannot solve the above problem, since it will significantly hurt the model performance for the remaining classes too. Therefore, we have also used other losses shown in eqs. 2, 3, 5, 6, 7, 8. In the case of the 3-way classifier example explained by the reviewer, the eq 1 of our proposed approach is not supposed to maintain the relationship between the logits of related classes. In fact, only applying eq 1 will significantly hurt the model performance, and the resulting model cannot achieve the goal of the 3-way classifier example. This is why our approach also uses other losses shown in eq 2, eq 5, eq 6. The L^{ne}\_c (eq 2) loss helps to maintain the performance of the model on the non-excluded classes. The knowledge-distillation based regularization loss $L^e_{kd}$ prevents the output logits corresponding to the non-excluded classes from changing for the images of the excluded  classes. Suppose, in the 3-way classifier example, if the output logit corresponding to the cat class for a leopard image was the second-highest, then after the deletion of the leopard class using our approach, the $L^e_{kd}$ loss ensures that the output logit corresponding to the cat class will now be the maximum for the leopard image. Therefore, our proposed approach ensures the requirement set by the reviewer. Further, if we wish to strictly enforce any relationship, e.g., all leopard images should be classified as a cat, then we can simply re-label the available limited leopard images as cat and apply the L^{ne}\_{c} loss for these examples using the new label to enforce this relationship in addition to the $L^e_{c}$ loss using the actual label to remove the excluded class information. Our proposed approach is flexible enough to incorporate these requirements. However, this is not the objective of our work.
> >
> > R2.3. The process of identifying parameters related to restricted classes seems quite empirically, as a transformation component is needed from some prior knowledge. The authors have mentioned it for images. However, many data privacy related data are also tabular. In this case, how to apply a proper transformation? If this component is quite related to data format, any workaround for this issue?
> >
> > Ans: Our proposed approach only requires the use of data augmentation/transformation techniques that are not used during training. Therefore, for any domain, we can use data augmentation methods compatible with the domain as long as they have not been used during training and do not change the label of the datapoint. We have provided a detailed discussion in point C2 of the common concerns regarding the effect of different data augmentations on the selection process of the restricted class relevant parameters.
> >
> >
> > R2.4. From Figure 3, KD is defined for remaining classes only, but the KD loss also includes restricted classes.
> >
> > Ans: We apologize for this confusion. However, Figure 3. is correct. We simply wanted to show that the KD loss is applied to only the logits corresponding to the remaining classes for the training examples from both excluded and non-excluded classes. We used curly brackets to highlight that the KD loss preserves the logits corresponding only to the non-excluded classes. We have also mentioned this point in Sec. 4 of the paper. We have further clarified this point in the revised paper by modifying Figure 3 to avoid any confusion.
> >
> >
> > R2.5. It is interesting to see the model performance comparison with the original training in term of remaining classes only (also related to concern 2, the model performance on the original raining data of remaining classes only may be a good reference point for evaluation), although original training data might be inaccessible in the proposed setting.
> >
> > Ans: We would like to point out that the model obtained after training on the entire training data of the remaining classes (FDR), is not a solution to our RCRMR-LD problem setting. Since only limited training data is available in our problem setting, the FDR model violates our problem setting and leads to an unfair comparison. All the compared baselines and methods in the tables follow the same setting. Therefore, we have not compared our approach with FDR in the tables provided in the paper.

---

> > > ### Author Response · Authors · 2021-11-28
> > > **Follow-up**
> > >
> > > Dear Reviewer,
> > >
> > > We are eagerly waiting for your comments regarding our response to your review, so that we can properly address any further concerns before the deadline for discussion that is very close.
> > >
> > > Regards,
> > >
> > > Authors

---

> > > ### Comment · Reviewer_3V1Q · 2021-11-30
> > > **Thanks for your response.**
> > >
> > > The authors have addressed part of my concerns.
> > >
> > > However, the concern 2 is actually regarding the evaluation of class removal. "making model forget some examples or some classes does not mean zero classification accuracy or random classification accuracy (i.e., 1/N)", so it is doubtful to use forgetting accuracy to evaluate the performance of class removal. Such concern is also related to the proposed concern 5.

---

> > > > ### Author Response · Authors · 2021-11-30
> > > > **Thank you for the response to our comments**
> > > >
> > > > Dear Reviewer,
> > > >
> > > > Thank you for your response. First of all, let us break down the statement “making model forget some examples or some classes does not mean zero classification accuracy or random classification accuracy” into two parts.
> > > >
> > > > Statement 1: “making model forget some examples does not mean zero classification accuracy or random classification accuracy”.
> > > >
> > > > Statement 2: “making model forget some classes does not mean zero classification accuracy or random classification accuracy”.
> > > >
> > > > If you use data deletion methods to make the model forget some examples, then statement 1 is true, since the objective of data deletion is to modify the model as if it was never trained on those examples [1,2]. Therefore, the classification accuracy for the class of those examples will still be decent since only some examples of that class have been forgotten. However, in our problem setting, we forget the entire class. Therefore, statement 1 is not relevant to our problem setting.
> > > >
> > > > The statement 2 is not valid. In fact [2] states that “for the case of removing an entire class
> > > > the target forget error (i.e. the error on the class to forget) is 100%”, i.e., 100-100 = 0% classification accuracy. [2] shows in Figure 6 (in the appendix) that their proposed method achieves this target using all the training data for a class. However, this approach cannot be applied to our problem setting since only very limited training data is available in the RCRMR-LD setting.
> > > >
> > > > [1] Making ai forget you: Data deletion in machine learning, Ginart et al., 2019.
> > > >
> > > > [2] Mixed-privacy forgetting in deep networks, Golatkar et al. CVPR 2021.
> > > >
> > > > Let us understand this point from the perspective of model optimization. When the model is untrained, then it has a random classification layer accuracy for any class. As we train the model using gradient descent on a loss function, the classification layer accuracy of the model improves from random. Therefore, if we now use gradient ascent to forget any class, the above process gets reversed and the classification layer accuracy falls (as shown in Figs. 5, 7 of the manuscript). Consequently, when the model forgets a class from all the layers, its classification layer accuracy for that class ($\texttt{FA}_\texttt{e}$) should return to random or zero.
> > > >
> > > > Therefore, a corrected statement should be the following.
> > > >
> > > > Statement 3: “Zero or random classification layer accuracy ($\texttt{FA}_\texttt{e}$) for a restricted class does not mean that the restricted class information has been removed from all layers of the model”
> > > >
> > > > In order to explain statement 3, we have already proposed the weight deletion WD baseline (baseline 1 in the manuscript) that simply deletes the classification layer weights corresponding to the restricted classes and achieves a classification layer accuracy ($\texttt{FA}_\texttt{e}$) of 0% for the restricted classes. However, this does not mean that the restricted class information has been removed from all the layers of the network. Since we are not aware of any other metric to measure the level of forgetting at the feature level, we propose a new metric referred to as forgetting prototype accuracy (explained in Sec. 1, 7). The forgetting prototype accuracy ($\texttt{FPA}_\texttt{e}$) of the WD deletion model is the same as that of the original model for the restricted classes. This clearly indicates that the restricted class information is still present in the layers of the network.
> > > >
> > > > Since we have already discussed this point in the paper (explained in Secs. 1, 8.1), we thought that your concern 2 was mainly about using data deletion, which we have explained in our response to R2.2 above, is not a solution to our RCRMR-LD problem. (contd.)

---

> > > > > ### Author Response · Authors · 2021-11-30
> > > > > **Thank you for the response to our comments (contd.)**
> > > > >
> > > > > In order to remove the restricted class information from all layers of the network, our proposed ERwP approach identifies the restricted class relevant parameters in all layers of the network and removes the restricted class information from them using gradient ascent.  Further, we also show that our proposed approach of identifying the restricted-class relevant parameters from which the restricted class information has to be removed is very effective (refer to our response to the point R3.5 of reviewer 3/pyXW). We do not specifically target the classification layer and in fact target all layers of the model. Still, our approach makes the classification layer forgetting accuracy ($\texttt{FA}_\texttt{e}$) almost zero for the restricted classes while maintaining the classification layer accuracy  ($\texttt{CA}_\texttt{ne}$) of the remaining classes (eqs. 2, 5, 6). Our approach also significantly reduces the forgetting prototype accuracy ($\texttt{FPA}_\texttt{e}$) of the restricted classes. This shows that our approach is very effective at removing the restricted class information from all layers of the model (as also pointed out by the other reviewers). Since our approach removes the restricted class information from all layers of the model, therefore, the resulting model achieves near-zero classification layer accuracy ($\texttt{FA}_\texttt{e}$) for the restricted classes and achieves a significantly lower forgetting prototype accuracy ($\texttt{FPA}_\texttt{e}$) of the restricted classes. Please also refer to our response to the point R4.2 of reviewer 4/Y25C, for a detailed discussion on the forgetting prototype accuracy.
> > > > >
> > > > > To provide a good reference for our forgetting prototype accuracy, we have shown that the forgetting prototype accuracy of the full data retraining (FDR) model (that has been trained on the entire training data of the remaining classes), is similar to the forgetting prototype accuracy of our approach (discussed in detail in the 2nd paragraph of Sec. 8.1). This demonstrates the effectiveness of our approach in removing restricted class information since the FDR model has not been trained on restricted classes. However, please note that comparing our model to the FDR model is unfair since it requires access to the full training data and is, therefore, not a solution to our proposed RCRMR-LD problem. Therefore, we have not provided these results in the table but only mentioned them in the text of Sec. 8.1.
> > > > >
> > > > > Regarding, concern 5, we cannot compare our approach with FDR in terms of the remaining classes, i.e., the constraint accuracy ($\texttt{CA}_\texttt{ne}$) of the remaining classes, since FDR is not a solution to our problem setting. We have instead clearly demonstrated in Table 1, 2, 6 that our approach maintains the constraint accuracy ($\texttt{CA}_\texttt{ne}$) of the remaining classes very close to that of the original pre-trained model, which is the objective of the RCRMR-LD problem setting.
> > > > >
> > > > > We have provided a detailed response to all your concerns. We believe these points address your concerns. Considering these points, will you kindly consider increasing your score since ICLR is a reputed conference that promotes new problem settings and ideas. Please update the rating in your review accordingly.
> > > > >
> > > > > Regards,
> > > > >
> > > > > Authors

---

> > > > > > ### Comment · Reviewer_3V1Q · 2021-12-01
> > > > > > **Thanks for your response.**
> > > > > >
> > > > > > I think "for the case of removing an entire class, the target forget error (i.e. the error on the class to forget) is 100%" is only valid when the restricted classes are irrelevant to the remaining classes; otherwise, the performance on the remaining classes would be degraded. So the forgetting accuracy may be not an appropriate evaluation metric in a general case of class removal.
> > > > > >
> > > > > > My concern 5 is actually to suggest to reference point for evaluation as you mentioned: "To provide a good reference for our forgetting prototype accuracy, we have shown that the forgetting prototype accuracy of the full data retraining (FDR) model (that has been trained on the entire training data of the remaining classes), is similar to the forgetting prototype accuracy of our approach".
> > > > > >
> > > > > > I raise the score but do not head for acceptance since the paper still needs a lot of effort to better merge all comments from reviewers after I checked the revision; in another hand, the proposed method also needs to be further improved especially the part of identifying the relevant
> > > > > > parameters for the restricted classes.

---

> > > > > > > ### Author Response · Authors · 2021-12-01
> > > > > > > **Thank you for the response to our comments**
> > > > > > >
> > > > > > > Dear Reviewer,
> > > > > > >
> > > > > > > Thank you for increasing your rating and for your response.
> > > > > > >
> > > > > > > We would like to point out that the knowledge-distillation based regularization loss (eq 5) used in our approach prevents the output logits corresponding to the non-excluded classes from changing for the images of the excluded classes as we perform gradient ascent on the images from the excluded classes. Therefore, even if the restricted classes are relevant to the remaining/non-excluded classes, this loss will ensure that the performance on the remaining classes is not degraded due to this dependency. Therefore, forgetting accuracy ($\texttt{FA}_\texttt{e}$) for the restricted classes is an appropriate evaluation metric for our approach. To verify whether that the model performance on the non-excluded classes is maintained, we also proposed a constraint accuracy metric ($\texttt{CA}_\texttt{ne}$) for the non-excluded classes. Therefore, as can be seen in Tables 1,2,6, our approach reduces the forgetting accuracy ($\texttt{FA}_\texttt{e}$) for the restricted classes close to zero while maintaining the constraint accuracy ($\texttt{CA}_\texttt{ne}$) for the non-excluded classes.
> > > > > > >
> > > > > > > We would also like to point out that we do not use a single metric to judge any approach and instead have proposed three novel performance metrics for this **new** RCRMR-LD problem setting in Sec.7 of the manuscript, i.e.,  forgetting accuracy ($\texttt{FA}_\texttt{e}$), forgetting prototype accuracy ($\texttt{FPA}_\texttt{e}$), and constraint accuracy ($\texttt{CA}_\texttt{ne}$). We have described these metrics in the manuscript in detail in Sec. 7. For the reviewer’s convenience, we have quoted the corresponding text. “The forgetting accuracy refers to the fully-connected classification layer accuracy of the model for the excluded classes. The forgetting prototype accuracy refers to the nearest class prototype-based classifier accuracy of the model for the excluded classes. $\texttt{CA}_\texttt{ne}$ refers to the fully-connected classification layer accuracy of the model for the non-excluded classes.”  Therefore, **using only a single metric will be misleading**.
> > > > > > >
> > > > > > > We have included all the changes suggested by the reviewers either in the main paper or in the appendix, which were found satisfactory by the other reviewers. The remaining points were mainly clarifications provided to the reviewers.
> > > > > > >
> > > > > > > We have also thoroughly revised the paper to include further details regarding the process for selecting the restricted-class relevant parameters as suggested by the reviewers. We have provided these additional details in Secs. 11.1 (PROCESS FOR SELECTING THE RESTRICTED CLASS RELEVANT PARAMETERS), 11.7 (EFFECT OF DIFFERENT DATA AUGMENTATIONS ON THE IDENTIFICATION OF CLASS RELEVANT MODEL PARAMETERS), and 11.8 (ABLATION EXPERIMENTS ON THE RESTRICTED CLASS RELEVANT PARAMETERS) in the appendix, due to the space constraint in the main paper. Therefore, we request you to kindly review these sections in the appendix, and we hope that this will address your concern.

---

### Official Review · Reviewer_yBwg · 2021-11-04

**Correctness:** 3
**Technical Novelty And Significance:** 2
**Empirical Novelty And Significance:** 2
**Recommendation:** 5
**Confidence:** 4

**Main Review:**

Positives:
1. The paper studies an important problem of tackling with restricted classes.
2. The presented approach displays an ability to remove restricted class information from model parameters.

Negatives:
1. The paper is not very clearly written, with concepts repeated several times and not clear description on some others that are mentioned below.
2. While the problem is interesting indeed, the motivation for the proposed solution is not clearly presented. Instead of repeating the ideas, it would be helpful to have a few clear examples that illustrate the need to solve this problem, as well as a clear description of the behavior of the said approach. While an example about the company logo is stated, it would be helpful to have a few more clear examples from real-world settings to help the reader. One such example, a model trained to predict which treatment would be beneficial for the patient would need to be altered if the treatment cannot be offered in the future due to ethical or resource constraints.
3. While empirical results on the CIFAR-100 and ImageNet-1K datasets seem promising, it would be helpful to study this in the real-world dataset. Issues such as generalizability due to distribution shifts in the future and fairness considerations when certain labels are dropped are potential directions.

Additional comments:
1. The notation for the excluded and non-excluded classes is a bit confusing as $C_e, C_r$ can both mean excluded or restricted. I would suggest to change this.

**Summary Of The Paper:**

The paper tackles the problem of restricted class unavailability after a deep learning model has already been trained on such restricted classes and the aim is to remove any information pertaining to the restricted classes from the model parameters so that the model will not be able to correctly classify the restricted classes in the future. The approach presented includes identifying the model parameters that are most relevant to the restricted classes and removing the restricted class information from these parameters (gradient ascent) while ensuring that these parameters can still be used for accurately classifying other non-restricted classes. With the need to correctly assess the utility of the proposed approach, several baseline methods have been proposed. Empirical results on the CIFAR-100 and ImageNet-1K datasets illustrate how the proposed approach can be used.

**Summary Of The Review:**

The paper studies an important problem. However, there are some challenges with respect to the writing, motivation of the solution and potentially several important directions that can be addressed.

---

> ### Author Response · Authors · 2021-11-23
> **Response to Reviewer 1 / yBwg**
>
> Thanks for your positive comments about our proposed problem setting and the effectiveness of our approach in removing the restricted class information from the model parameters. We have responded to your concerns below. We sincerely hope our response will help you rethink your assessment. We have thoroughly updated the paper to address your concerns. Please update your review accordingly.
>
> R1.1. The paper is not very clearly written, with concepts repeated several times and not clear description on some others that are mentioned below.
>
> Ans: We apologize if some of the details were not clear. We have clarified these points below and have included these points in the revised version for improved readability.
>
> R1.2. While the problem is interesting indeed, the motivation for the proposed solution is not clearly presented. Instead of repeating the ideas, it would be helpful to have a few clear examples that illustrate the need to solve this problem, as well as a clear description of the behavior of the said approach. While an example about the company logo is stated, it would be helpful to have a few more clear examples from real-world settings to help the reader. One such example, a model trained to predict which treatment would be beneficial for the patient would need to be altered if the treatment cannot be offered in the future due to ethical or resource constraints.
>
> Ans: Thank you for your suggestions. We have provided some real-world examples that can suffer from the proposed RCRMR-LD problem in point C1 of the common concerns, which serve as motivations for the proposed problem setting. We have included these examples in the revised paper in Secs. 1, 3. We have also included the treatment prediction example in our revised paper, as suggested by the reviewer.
>
> R1.3. While empirical results on the CIFAR-100 and ImageNet-1K datasets seem promising, it would be helpful to study this in the real-world dataset. Issues such as generalizability due to distribution shifts in the future and fairness considerations when certain labels are dropped are potential directions.
>
> Ans: The ImageNet-1K is a vast dataset containing real-world images and is extensively used in the computer vision community to train or pre-train deep learning models. CUB-200 dataset is also a real-world dataset containing images of birds, and we have provided experimental results for CUB-200 in the appendix. We have now also described in Sec.3 of the revised paper a real-world setting, i.e., incremental learning, which can suffer from our proposed RCRMR-LD problem. The incremental learning problem is a practical, real-world setting where new classes become incrementally available for training. We have also described how our approach can address the RCRMR-LD problem even in the incremental learning setting and have provided experimental results in Table 4 in the revised paper to validate our approach in this setting. Therefore, we have demonstrated that the RCRMR-LD problem can occur in real-world problem settings, and our approach can effectively address this problem in those settings.
>
>
> The model achieved after applying our proposed approach has almost the same test data generalization power as the original model, as can be seen from their performance on the same test images in Tables 1, 2, 5. The deletion of any class may, in very rare cases, lead to a fairness/bias problem. However, this is a data-based problem, and any model will suffer from bias if the data is biased unless we apply any bias mitigation approach. We believe that this direction of work is different from the direction of our work in this paper. Part of your concern has been addressed in our response to the concern R2.2. of reviewer R2/3V1Q (refer to the second paragraph of the response).
>
> R1.4. The notation for the excluded and non-excluded classes is a bit confusing as $C_e$, $C_r$ can both mean excluded or restricted. I would suggest to change this.
>
>  Ans: Thank you for pointing this out. We have now revised the notations in the revised paper to $C_e$ and $C_{ne}$ that refer to the excluded and non-excluded classes, respectively.

---

> > ### Author Response · Authors · 2021-11-28
> > **Follow-up**
> >
> > Dear Reviewer,
> >
> > We are eagerly waiting for your comments regarding our response to your review, so that we can properly address any further concerns before the deadline for discussion that is very close.
> >
> > Regards,
> >
> > Authors

---

> > > ### Comment · Reviewer_yBwg · 2021-11-29
> > > **Thank you for the response**
> > >
> > > I thank the authors for the clarifications. I have carefully read the authors' responses as well as other comments. The authors have addressed most of my concerns, I keep my score but believe that the problem is interesting and the work paves an important direction.

---

> > > > ### Author Response · Authors · 2021-11-29
> > > > **Thank you for your response**
> > > >
> > > > Dear Reviewer,
> > > >
> > > > We thank you for the time invested by you in carefully reading our revised paper and responses. Your comments helped us to improve our paper. We have addressed your concerns, and you have already mentioned, our paper is interesting, and our work paves an important direction. You have also provided a positive feedback on the ability of our method to remove restricted class information from model parameters. Further, we also propose a novel approach for identifying the restricted class-relevant parameters. Considering these points, will you kindly consider increasing your score since ICLR is a reputed conference that promotes new problem settings and ideas. Please update the rating in your review accordingly.
> > > >
> > > > Regards,
> > > >
> > > > Authors

---

### Author Response · Authors · 2021-11-23
**Common Concerns**

We appreciate the time invested by the reviewers in carefully reading our paper and providing very helpful comments. We sincerely thank the reviewers for their positive feedback on: the novelty of the problem setting [R1/yBwg, R3/pyXW, R4/Y25C], extensive evaluation [R3/pyXW,R4/Y25C], superior performance of proposed approach [R1/yBwg, R2/3V1Q, R3/pyXW, R4/Y25C].

C1. Motivation for our Proposed Problem Setting

We explain below, how existing problem settings may face the problem of restricted category removal and describe how our approach can address them. We have included these discussions in Secs.~1, 3 of our revised paper in order to clarify the motivation for our problem setting. We have also included the treatment prediction example suggested by a reviewer in our revised paper. We provide the following specific scenarios to demonstrate that our proposed problem can occur in real-world scenarios.

A real-world scenario where our proposed problem can arise is federated learning [1]. In the federated learning setting, there are multiple collaborators that have a part of the training data stored locally, and a model is trained collaboratively using these private data without sharing or collating the data due to privacy concerns. Suppose organization A has a part of the training data, and there are other collaborators that have other parts of the training data for the same classes. Organization A collaboratively trains a model with other collaborators using federated learning. After the model has been trained, a few classes may become restricted in the future due to some ethical or privacy concerns, and these classes should be removed from the model. However, the other collaborators may not be available or may charge a huge amount of money for collaborating again to train a fresh model from scratch. In this case, organization A does not have access to the full training data of the non-excluded/remaining classes that it can use to re-train a model from scratch in order to exclude the restricted classes information. This scenario clearly demonstrates our proposed problem setting. In such a case, our approach provides a feasible solution to optimize the model on the limited training data available with organization A to remove the restricted classes from the model.

Another real-world example of our proposed problem is the MS-Celeb-1M [2] dataset. This dataset contained the face images of 1 million celebrities, and the identity of the individual served as the class label. Researchers have used this dataset extensively to address many types of computer vision problems. However, it was later found that many of the individuals in the dataset were not celebrities but professionals who had not provided any consent for the use of their photographs. Consequently, this dataset was retracted due to privacy concerns. A similar problem occurred with the DukeMTMC [3] dataset. The models trained on MS-Celeb-1M contain the information regarding the restricted classes. Since the original dataset is not available, it is highly possible that only a few images of these non-celebrity individuals (restricted classes) are available on the internet. In such a situation, our proposed approach provides an efficient approach of removing the information of these classes from the parameters of the trained model.

Another real-world scenario is the incremental learning setting [4,5], where the model receives training data in the form of sequentially arriving tasks. Each task contains a new set of classes. During a training session $t$, the model receives the task $t$ for training and cannot access the full data of the previous tasks. Instead, the model has access to very few exemplars of the classes in the previous tasks. Suppose before training a model on training session $t$, it is noticed that some classes from a previous task ($<t$) have to be removed from the model since those classes have become restricted due to privacy or ethical concerns. In this case, only a limited number of exemplars are available for all these previous classes (restricted and remaining). Therefore, our approach can be used to optimize the model in order to remove the information regarding the restricted classes without affecting the model performance on the remaining classes. Further, our approach will be able to achieve this objective using the limited number of exemplars available for these classes.


[1]  Communication-Efficient Learning of Deep Networks from Decentralized Data, Brendan McMahan et al., ICAIS 2017

[2]  Ms-celeb-1m: A dataset and benchmark for large-scale face recognition, Guo et al., ECCV 2016

[3]  Performance measures and a data set for multi-target, multi-camera tracking, Ristani et al., ECCV 2016

[4]  Icarl: Incremental classifier and representation learning, Rebuffi et al., CVPR 2017

[5]  Fearnet: Brain-inspired model for incremental learning, Kemker et al., ICLR 2018

---

> ### Author Response · Authors · 2021-11-23
> **Common Concerns (contd.)**
>
> C1 (contd.)
>
> We will now experimentally demonstrate how the RCRMR-LD problem in the incremental learning setting is addressed using our proposed approach. We consider an incremental learning setting on the CIFAR-100 dataset in which each task contains 20 classes. We use the BIC [6] method for incremental learning on this dataset. The exemplar memory size is fixed at 2000 as per the setting in [6]. In this setting, there are 5 tasks, and task t becomes available for training during training session t. Let us assume that the model has already been trained on 4 tasks (80 classes), and we are in the fifth training session. Suppose, at this stage, it is noticed that all the classes in the first task (20 classes) have become restricted and need to be removed before the model is trained on task 5. However, we only have a limited number of exemplars of the 80 classes seen till now, i.e., 2000/80 = 25 per class. Directly training the model on this limited data for the remaining classes will lead to very poor performance. This is, therefore, a practical example of our proposed problem setting. We apply our proposed approach to the model obtained after training session 4, and the results are reported in Table~4 of the revised paper. The results indicate that our approach modified the model obtained after session 4, such that the forgetting accuracy of the restricted classes approaches 0% and the constraint accuracy of the remaining classes is not affected. In fact, the modified model behaves as if, it was never trained on the classes from task 1. After this process, we can continue the incremental training of the modified model on the training data from task 5.
>
> The examples mentioned above show that our proposed RCRMR-LD problem is practically possible in real world scenarios. This forms the basis of the motivation to address our proposed RCRMR-LD problem setting.
>
> We have proposed a new problem setting that can occur in various real-world scenarios. We have explained some of the examples of this setting above. As issues of privacy increase, more such problems can arise in the future. ICLR is a reputed conference that promotes new problem settings and approaches in order to address problems that may arise in the future. Therefore, we believe that our work should be encouraged.
>
> [6] Largescale incremental learning, Wu et al., CVPR 2019

---

> > ### Author Response · Authors · 2021-11-23
> > **Common Concerns (contd.)**
> >
> > C2. Clarification Regarding the Transformations/Data-Augmentation used in our Approach
> >
> > We apologize for the confusion regarding the data augmentation/transformation techniques used in our approach for identifying the class-relevant parameters of the model for any given restricted class. Basically, the purpose of applying any data augmentation (not used during training) in our approach is to study the gradient updates when the model performs backpropagation over slightly different versions of the training data of a class and use this information to identify the highly relevant parameters of the model with respect to that class. We have performed experiments using various data augmentation techniques and have provided these results in Fig-6 of Sec.-11.7 in the appendix of the revised paper. The results from these experiments indicated that the data augmentation techniques are almost equally effective in our approach for finding the relevant parameters with respect to any restricted class. For any domain, we can use the data augmentation methods compatible with that domain as long as they have not been used during training and do not change the label of the datapoint. This holds true even for the data augmentation techniques used for other types of data, such as tabular/categorical data and those used for NLP tasks.

---

> > > ### Author Response · Authors · 2021-11-23
> > > **Common Concerns (contd.)**
> > >
> > > In this work, we have proposed a new problem setting, created various baselines, proposed performance metrics for this setting, and proposed a novel method to address this problem. In order to improve the paper, we have also added further discussions regarding the motivations of the problem in the revised paper. It is not possible to retain all the details of our work in the main paper due to lack of space. Therefore, we have shifted the detailed discussion on the baselines to the appendix. We hope that this change is not taken negatively.

---

### Author Response · Authors · 2021-11-26
**Follow-up**

Dear Reviewers,

Did you get a chance to read our responses to your comments? Since the deadline for discussion is approaching very fast, we would like to know your opinion regarding our responses, so that we can properly clarify any further concerns.

Regards,

Authors

---

### Author Response · Authors · 2021-11-30
**Change-Log**

Dear Reviewers and AC,

We have thoroughly revised the manuscript to incorporate the suggestions of the reviewers. The main changes are as follows:

1. We have revised Sec. 1 to include the treatment prediction example as suggested by reviewer 1/yBwg.
2. We have added a new section Sec. 3 - “RCRMR-LD PROBLEM IN REAL WORLD SCENARIOS” to explain the real-world scenarios where our proposed RCRMR-LD problem can occur as suggested by all reviewers.
3. We have added a new section Sec. 8.3 - “RCRMR-LD PROBLEM IN INCREMENTAL LEARNING” to demonstrate the RCRMR-LD problem in an incremental learning setting. We have added a new Table 4 to empirically show that our proposed ERwP approach addresses the RCRMR-LD problem in the incremental learning setting.
4. We have thoroughly revised the notation throughout the manuscript for improved readability as suggested by reviewer 1/yBwg.
5. We have modified Fig. 3 to avoid any confusion as suggested by reviewer 2/3V1Q.
6. We have added a new section Sec. 11.1 - “PROCESS FOR SELECTING THE RESTRICTED CLASS RELEVANT PARAMETERS” in the appendix to provide a detailed description of the process for identifying the restricted class relevant parameters as suggested by reviewer 3/pyXW.
7. We have added a new section Sec. 11.7 - “EFFECT OF DIFFERENT DATA AUGMENTATIONS ON THE IDENTIFICATION OF CLASS RELEVANT MODEL PARAMETERS” and Fig. 6 in the appendix to experimentally demonstrate the effect of using different data augmentation techniques on our approach for identifying the restricted class-relevant parameters as suggested by reviewers 3V1Q, pyXW, Y25C.
8. We have added a new section Sec. 11.8 - “ABLATION EXPERIMENTS ON THE RESTRICTED CLASS RELEVANT PARAMETERS” in the appendix to experimentally demonstrate that our proposed approach of identifying the restricted class relevant parameters is very effective as suggested by reviewer 3/pyXW.

Regards,

Authors

---

### Decision · Program_Chairs · 2022-01-20

**Decision:**

Reject

**Comment:**

The paper proposes a technique to efficiently retrain a model when a small number of classes are required to be removed.
Reviewers in general like the paper, but the key issue is motivation for the problem. The motivating examples in the rebuttal are not very good because a. authors do not provide any evidence that such situations are critical or commonplace, b. the data points that are available for retraining might be very biased.
A more careful grounding of the work would be important to motivate the ICLR community and the ML community in general to further study this problem. But for now, unfortunately the paper does not seem ready for publication at ICLR.